# Inverse Reinforcement Learning of Interactive Scenarios

## Abstract

This paper studies the problem where a learner aims to learn the reward function of an expert and a policy to interact with the expert from interactions with the expert. We formulate the problem as a stochastic bi-level optimization problem where the lower level learns a reward function that explains the behaviors of the expert, and the upper level learns a policy to interact with the expert. We develop a double-loop algorithm, General Scenario Interactive Inverse Reinforcement Learning (GSIIRL), which solves the lower-level optimization problem in the inner loop and the upper-level optimization problem in the outer loop. We formally guarantee that GSIIRL converges at the rate of $\mathcal{O}(1/\sqrt{K})$ and empirically validate our algorithm through simulations.

## 1 Introduction

Inverse reinforcement learning (IRL) aims to recover a reward function and a corresponding policy that best explain an expert's behavior given the expert's demonstration trajectories. IRL has been applied across diverse domains, including robotics (Ziebart et al., 2008; Okal & Arras, 2016), cybersecurity (Zhang et al., 2019; Elnaggar & Bezzo, 2018), and biology (Hirakawa et al., 2018; Ashwood et al., 2022). In standard IRL, the expert's demonstration trajectories are assumed to be fixed and unaffected by the learner, as the learner passively observes the expert and learns from these demonstrations without influencing the expert's behavior. However, recent applications (Palaniappan et al., 2017; Büning et al., 2022; Kamalaruban et al., 2019) have motivated an interactive form of IRL where the learner actively interacts with the expert to infer the reward function and policy through these interactions. For example, a robot needs to infer a human's intended destination in a maze and then open the appropriate doors to help the human reach that goal (Büning et al., 2022). In this scenario, the learner (robot)'s actions (e.g., opening different doors) directly influence the expert (human)'s trajectory, violating the standard IRL assumption of passive observation and rendering conventional IRL methods inapplicable.

To bridge this gap, this paper studies interactive IRL, where a learner interacts with an expert to learn a reward function that explains the expert's behavior (i.e., such that the expert's trajectories are optimal under the learned reward) and a policy for effectively interacting with the expert. In this interactive setting, the learner is not merely an observer but an active participant: by interacting with the expert, the learner can influence the expert's trajectories. Consequently, if the learner's policy changes, the expert's trajectories may change accordingly. Consider a scenario where a human (expert) and a ground mobile robot (learner) share a 2D environment while navigating to their respective destinations. The human has the right-of-way, so the robot must either slow down or detour to allow the human to cross first. At the same time, the robot is trying to reach its own destination as soon as possible. Thus, the robot has a human-independent objective (e.g., reaching its destination) as well as a human-dependent objective (e.g., bypassing the human). Accordingly, the robot's reward function can be decomposed into a human-independent part and a human-dependent part. For example, in (El-Shamouty et al., 2020), the reward function is defined by two terms: the robot's distance to its goal and whether a collision with the human occurs.

Motivated by this example, this paper assumes that the learner's reward function can be decomposed into an expert-independent component and an expert-dependent component. In our interactive IRL framework, the relationship between the learner's and the expert's reward functions is more general than those in fully cooperative interactive IRL (Palaniappan et al., 2017; Büning et al., 2022; Ka-

malaruban et al., 2019; Hadfield-Menell et al., 2016) and fully competitive interactive IRL (Zhang et al., 2019; Wang & Klabjan, 2018) settings. In the fully cooperative case, the learner and the expert share an identical reward function, whereas in the fully competitive case, the learner's reward function is the negation of the expert's reward function.

The interactive IRL problem of interest is closely related to RL (Lowe et al., 2017; Yu et al., 2022; Kuba et al., 2021), and classic IRL (Lin et al., 2019; Yu et al., 2019; Liu & Zhu, 2022). However, standard RL algorithms are not applicable to our setting because they require access to state-action-reward tuples of the expert, which are unavailable in interactive IRL. Similarly, classical IRL algorithms require passive observation, whereas in our setting, the learner actively interacts with the expert. These limitations also apply to multi-agent RL (MARL) and multi-agent IRL (MA-IRL), respectively. Detailed comparisons will be further elaborated in Section 3.

**Contribution.** This paper makes four main contributions. First, we study an interactive IRL problem, where the learner learns a reward function to explain the expert's behaviors and a policy to interact with the expert. We formulate this as a stochastic bi-level optimization problem in which the lower level learns a reward function to explain the expert's behavior and the upper level learns a policy to interact with the expert. Second, we develop a double-loop algorithm, General Scenario Interactive Inverse Reinforcement Learning (GSIIRL), to solve the bi-level optimization problem, with an outer loop handling the upper-level optimization problem and an inner loop handling the lower-level optimization problem. A key challenge in this approach is computing the hypergradient (the gradient of the upper-level objective function), since it involves an intractable Hessian. We address this by using simultaneous perturbation stochastic approximation (SPSA) (Spall, 1992) to approximate the hypergradient, reducing the computational complexity per outer-loop iteration from $\mathcal{O}(T^2)$ to $\mathcal{O}(T)$, where $T$ is the maximum trajectory length. Third, we theoretically guarantee that the algorithm converges at a rate of $\mathcal{O}(1/\sqrt{K})$, where $K$ is the number of outer-loop iterations. Fourth, we validate our algorithm through four experiments and show that GSIIRL performs comparably to benchmark methods (MARL and MA-IRL) that require additional information (either the expert's reward function or learner's optimal demonstrations).

## 2 RELATED WORK

**IRL and multiagent IRL (MA-IRL).** As (Ng & Russell, 2000) mentioned, IRL faces the challenge that the demonstrated trajectories can be explained by multiple reward functions. Several approaches have been proposed to address this challenge. The current state-of-the-art IRL methods include maximum margin IRL (Ng & Russell, 2000; Abbeel & Ng, 2004), maximum entropy IRL (Ziebart et al., 2008; 2010), maximum likelihood IRL (Zeng et al., 2022; 2023), and Bayesian IRL (Ramachandran & Amir, 2007; Choi & Kim, 2012). However, these methods all focus on single-expert scenarios. MA-IRL extends IRL to settings with multiple experts, where one or more learners recover the reward functions of multiple experts from their demonstrations (Lin et al., 2019; Yu et al., 2019; Liu & Zhu, 2022). These MA-IRL approaches still assume that the learners are isolated from the experts (i.e., no direct interaction between them) and thus cannot address interactive IRL scenarios.

**Fully cooperative and fully competitive interactive IRL.** Prior work on interactive IRL has primarily explored two cases: fully cooperative and fully competitive scenarios. In the fully cooperative setting, the learner and expert share the same reward function, and their interaction is formulated as a cooperative game (Palaniappan et al., 2017; Büning et al., 2022; Kamalaruban et al., 2019; Hadfield-Menell et al., 2016). In this case, the learner's objective is to recover the expert's reward function. In contrast, the fully competitive setting assumes the learner's reward is the negative of the expert's reward, leading to a zero-sum game (Zhang et al., 2019; Wang & Klabjan, 2018). Here, the learner aims to learn an objective that directly opposes the expert's reward. Both of these approaches impose strong assumptions on the relationship between the expert's and learner's reward functions, specifically assuming they are either identical or exact opposites. In this paper, the relationship between the reward functions of the learner and the expert is arbitrary.

**Bi-level optimization.** Bi-level optimization has been applied to many machine learning problems, including meta-learning (Lee et al., 2019; Xu & Zhu, 2022), hyperparameter optimization (Pedregosa, 2016; Xu & Zhu, 2023), and IRL (Liu & Zhu, 2022; 2024). A classic approach to such problems is the descent method (Kolstad & Lasdon, 1990; Xu & Zhu, 2023). This method typically requires computing the second-order Hessian of the lower-level objective function, an operation that

is prohibitively expensive to compute in our setting. A common strategy to avoid explicitly computing the Hessian is to approximate it using finite differences (Strikwerda, 2004). In this paper, we further reduce the computational burden by adopting the SPSA method, which perturbs all dimensions of the decision variable simultaneously and therefore requires only two objective function evaluations per iteration.

## 3 Model and problem statement

In this section, we introduce the interactive IRL problem.

**Markov Game model.** We model the interactions between a learner and an expert as a finite horizon Markov Game (MG) $(S, A, P, T, r_{ld} + r_{li}, r_e, \gamma)$. The elements of the MG are defined as follows:

- $S \triangleq S_l \times S_e$ is the continuous state space where $S_l$ and $S_e$ are the state spaces of the learner and the expert, respectively. We denote $s = (s_l, s_e) \in S$, with $s_l \in S_l$ and $s_e \in S_e$.
- $A \triangleq A_l \times A_e$ is the joint action space, where $A_l$ and $A_e$ are the continuous action spaces of the learner and the expert, respectively. We denote $a = (a_l, a_e) \in A$, with $a_l \in A_l$ and $a_e \in A_e$.
- $P(s'|s, a)$ is the transition probability density for moving from state $s$ to $s'$ by taking joint action $a$.
- $T$ is the finite time horizon.
- $r_{ld} + r_{li}$ is the reward function of the learner, consisting of two parts: the expert-dependent reward function $r_{ld}$ and the expert-independent reward function $r_{li}$. The function $r_{ld}$ maps full state-action pairs $(s, a)$ to bounded rewards, and $r_{li}$ maps learner-only state-action pairs $(s_l, a_l)$ to bounded rewards.
- $r_e$ is the expert's reward function, mapping state-action pairs $(s, a)$ to bounded rewards.
- $\gamma \in (0, 1]$ is the discount factor.

We define $\pi_l(a_l|s)$ as the learner's policy and $\pi_e(a_e|s)$ as the expert's policy. The joint policy is defined as $\pi(a|s) \triangleq \pi_l(a_l|s) \times \pi_e(a_e|s)$, which represents the probability density of the learner taking action $a_l$ and the expert taking $a_e$ at state $s$. When the joint policy $\pi$ is executed, the MG generates a trajectory $\zeta = s^0, a^0, s^1, a^1, \cdots, s^{(T-1)}, a^{(T-1)}$.

The learner's policy $\pi_l$ aims to maximize the cumulative reward $E^{\pi_l, \pi_e}[\sum_{t=0}^{T-1} \gamma^t (r_{ld}(s^t, a^t) + r_{li}(s_l^t, a_l^t))]$. Analogously, the expert's policy $\pi_e$ aims to maximize $E^{\pi_l, \pi_e}[\sum_{t=0}^{T-1} \gamma^t r_e(s^t, a^t)]$.

**Knowledge and goal of the learner.** The learner does not know the expert's reward function $r_e$ but knows its own expert-independent reward function $r_{li}$. It can observe state-action pairs $(s, a)$ and interact with the expert. Based on the trajectories collected during the interactions, the learner aims to recover the expert's reward function $r_e$ and compute the optimal joint policy $\pi^*$.

**Benefits of learning reward functions.** As noted by the seminal work (Ng & Russell, 2000), "the reward function is the most succinct, robust, and transferable definition of the task". When the transition probabilities of a Markov Game substantially change, the policy needs to be retrained, but the underlying reward function can remain the same. Furthermore, learning a human's reward function can facilitate the robot's policy learning (Büning et al., 2022). If the ground-truth $r_{ld}$ and $r_e$ are sparse, learning dense reward functions to approximate them can accelerate the policy learning while leaving the optimal policy unchanged (Memarian et al., 2021). Given the importance of reward functions, we include learning $r_e$ as a central goal for the learner.

We next discuss the distinctions between interactive IRL and related problems, including RL, IRL, fully cooperative interactive IRL, and fully competitive interactive IRL.

**Distinctions from RL.** In our interactive IRL setting, the learner's objective explicitly includes learning the expert's reward function $r_e$ from observed expert trajectories. These trajectories contain implicit information about $r_e$ because they are generated by the expert's policy $\pi_e$, which maximizes the expected cumulative reward $E^{\pi_l, \pi_e}[\sum_{t=0}^{T-1} \gamma^t r_e(s^t, a^t)]$. Model-free RL algorithms are not designed to learn reward functions. Model-based RL algorithms, on the other hand, can learn the expert's reward function given complete state-action-reward tuples. However, in our case, the learner only observes state-action pairs without access to the expert's reward signals, making model-based RL approaches also inapplicable. Similarly, MARL algorithms, as extensions of standard RL to multi-agent settings, inherit this limitation and thus cannot address interactive IRL problems.

**Distinctions from IRL.** In classical IRL, the learner passively observes expert demonstrations and has no influence over the expert's behavior. The expert's policy $\pi'_e$ results from maximizing the expected cumulative reward $E^{\pi'_e}[\sum_{t=0}^{T-1} \gamma^t r_e(s_e^t, a_e^t)]$. Specifically, the expert's action depends solely on its own state, and its trajectories are determined entirely by its policy and the environment dynamics. In contrast, interactive IRL is formulated within an MG framework, where the learner and expert interact with each other. In this setting, the expert's actions depend on the joint state of both the expert and the learner. Thus, the learner's policy influences its own state, which in turn affects the joint state, thereby affecting the expert's actions and resulting trajectories. Here, the expert's policy $\pi_e$ maximizes the expected cumulative reward $E^{\pi_l,\pi_e}[\sum_{t=0}^{T-1} \gamma^t r_e(s^t, a^t)]$, explicitly incorporating the learner's policy $\pi_l$. Due to these fundamental differences, classical IRL algorithms are unsuitable for interactive IRL problems, including ours, fully cooperative IRL (Palaniappan et al., 2017; Büning et al., 2022; Kamalaruban et al., 2019; Hadfield-Menell et al., 2016) and fully competitive IRL (Zhang et al., 2019; Wang & Klabjan, 2018). MA-IRL, as an extension of IRL to multi-agent settings, inherently shares this limitation and thus cannot address interactive IRL problems.

**Distinctions from fully cooperative and fully competitive interactive IRL.** Fully cooperative interactive IRL settings (Palaniappan et al., 2017; Büning et al., 2022; Kamalaruban et al., 2019; Hadfield-Menell et al., 2016) assume that the learner's and expert's reward functions are identical. Fully competitive interactive IRL settings (Zhang et al., 2019; Wang & Klabjan, 2018) assume that the learner's reward function is the negative of the expert's reward function. In our interactive IRL setting, the relationship between the learner's and the expert's reward functions can be arbitrary, which encompasses both identical and opposite cases.

## 4 PROBLEM FORMULATION AND BI-LEVEL SETUP

In this section, we formulate the learning problem in Section 3 as a bi-level optimization problem where the lower-level optimization problem learns the expert reward function $r_e$ and the upper-level optimization problem learns the optimal joint policy $\pi^*$.

Recall that $r_e$ and $r_{ld}$ are unknown to the learner. The learner uses a parametric function $r_{\theta_e}$ to estimate $r_e$ and uses $r_{\theta_l}$ to estimate $r_{ld}$ where $\theta_e, \theta_l \in \Theta \triangleq \{\theta | \|\theta\|_2 \leq 1\}$.

We define the policy $\pi_{\theta_l,r_e}$ as the optimal joint policy when the learner's reward function is $r_{\theta_l} + r_{li}$ and the expert's reward function is $r_e$. Analogously, given reward functions $r_{\theta_l} + r_{li}$ and $r_{\theta_e}$ for the learner and the expert respectively, the optimal joint policy is $\pi_{\theta_l,\theta_e}$.

**Learning expert's reward function $r_e$.** The learner aims to infer the expert's reward function according to the given estimate $r_{\theta_l}$. During interactions between the learner (using $r_{\theta_l}$) and the expert (using the ground-truth reward function $r_e$), the learner collects a set of trajectories $D_{\theta_l,r_e} \triangleq \{\zeta_i\}_{i=1}^d$ generated by the policy $\pi_{\theta_l,r_e}$. Since the expert utilizes the ground-truth reward function $r_e$, the trajectories recorded from the interactions are the demonstrations of the expert. Given that the learner knows $r_{\theta_l}$, inferring $r_e$ from collected demonstrations is a special case of the standard two-agent IRL problem (Lin et al., 2019; Yu et al., 2019; Liu & Zhu, 2022) with one of the reward functions

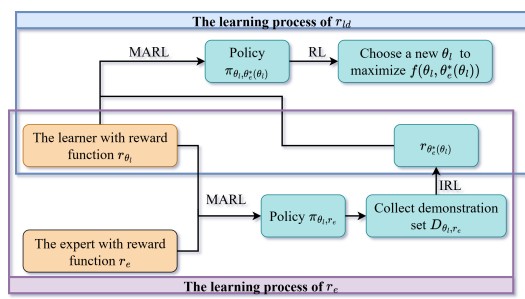

Figure 1: Flowchart of the overall learning process. The process of learning $r_{ld}$ is included in the upper block, and the process of learning $r_e$ is included in the lower block.

known to the learner. This learning procedure, depicted in the upper block of Figure 1, can be formulated as the following maximum likelihood IRL (ML-IRL) problem:

$$\theta_e^*(\theta_l) = \underset{\theta_e \in \Theta}{\arg\min} \quad L(\theta_l, \theta_e), \tag{1}$$

where the loss function is $L(\theta_l, \theta_e) \triangleq -\sum_{i=1}^d \sum_{t=0}^{T-1} [\ln \pi_{\theta_l,\theta_e}(a^{it}|s^{it})] + \frac{\lambda}{2}\|\theta_e\|_2^2$, $(a^{it}, s^{it}) \in \zeta_i \in D_{\theta_l,r_e}$. This ML-IRL approach seeks a reward parameter $\theta_e^*(\theta_l)$ such that the associated reward function $r_{\theta_e^*(\theta_l)}$ maximizes the likelihood of generating the observed trajectories in $D_{\theta_l,r_e}$. With a

given $r_{\theta_l}$, the policy $\pi_{\theta_l, \theta_e^*(\theta_l)}$ induced by the learned expert's reward function $r_{\theta_e^*(\theta_l)}$ is expected to produce the same trajectory from the policy $\pi_{\theta_l, r_e}$. Notice that the maximum likelihood estimation has been extensively applied in IRL (Ziebart et al., 2008; 2010). Additionally, to promote simpler reward structures, an $L_2$ regularization term $\frac{\lambda}{2}\|\theta_e\|_2^2$ with a small positive scalar $\lambda$ is incorporated.

**Learning the joint policy $\pi^*$ and the expert-dependent reward function $r_{ld}$.** Given a learner's reward parameter $\theta_l$, the learner solves problem (1) to obtain the expert's reward parameter $\theta_e^*(\theta_l)$ and the corresponding policy $\pi_{\theta_l, \theta_e^*(\theta_l)}$. The learner's objective is to learn a parameter $\theta_l$ such that the policy $\pi_{\theta_l, \theta_e^*(\theta_l)}$ maximizes the cumulative reward $E^{\pi_{\theta_l, \theta_e^*(\theta_l)}}[\sum_{t=0}^{T-1} \gamma^t(r_{ld}(s^t, a^t) + r_{li}(s_l^t, a_l^t))]$. This learning process is depicted in the lower block of Figure 1 and formulated as follows:

$$\underset{\theta_l \in \Theta}{\arg\min} \quad f(\theta_l, \theta_e^*(\theta_l)) \triangleq -E^{\pi_{\theta_l, \theta_e^*(\theta_l)}}\Big[\sum_{t=0}^{T-1} \gamma^t(r_{ld}(s^t, a^t) + r_{li}(s_l^t, a_l^t))\Big] \tag{2}$$

To solve the problem (2), the learner needs to interact with the expert and receive the reward values of $r_{ld}(s^t, a^t)$. The learner updates the parameter $\theta_l$ via policy gradient by computing the gradient $\nabla_{\theta_l} f(\theta_l, \theta_e^*(\theta_l))$, as the policy itself is parameterized by $\theta_l$.

**Overall learning process.** The entire learning procedure exhibits a hierarchical structure. Given the current learner's reward parameter $\theta_l$, the lower level solves the ML-IRL problem (1) to determine the expert's reward function $r_{\theta_e^*(\theta_l)}$. The upper level optimizes the expert-dependent reward function $r_{\theta_l}$ to solve the RL problem (2). This hierarchical optimization framework is illustrated in Figure 1 and can be formulated as a bi-level optimization problem:

$$\underset{\theta_l \in \Theta}{\arg\min} \quad f(\theta_l, \theta_e^*(\theta_l)), \quad \text{s.t.} \quad \theta_e^*(\theta_l) = \underset{\theta_e \in \Theta}{\arg\min} \quad L(\theta_l, \theta_e). \tag{3}$$

Once $r_{\theta_e^*(\theta_l^*)}$ and $r_{\theta_l^*}$ are determined, the joint optimal policy $\pi_{\theta_l^*, \theta_e^*(\theta_l^*)}$ is readily computed.

## 5 ALGORITHM

In this section, we develop a double-loop algorithm named GSIIRL (Algorithm 1) to solve the bi-level optimization problem described in (3). In each $k$-th outer loop iteration, the learner partially solves the lower-level optimization problem within an inner loop and subsequently employs this intermediate solution to tackle the upper-level optimization problem through an outer loop. The specifics of the inner and outer loops are detailed in Sections 5.1 and 5.2, respectively.

### 5.1 INNER LOOP

At each iteration $t$ of the inner loop, the learner updates the parameter $\theta_e(t)$ via projected SGD. Specifically, $\theta_e(t)$ is updated by moving in the opposite direction of the partial gradient $\nabla_{\theta_e} L(\theta_l(k), \theta_e(t))$ and projecting the resulting parameter back onto the feasible set $\Theta$. The analytical expression of the gradient $\nabla_{\theta_e} L(\theta_l, \theta_e)$ is provided in the following lemma.

**Lemma 1.** *The gradient* $\nabla_{\theta_e} L(\theta_l, \theta_e) = \mu_e(\pi_{\theta_l, \theta_e}) - \hat{\mu}_e(D_{\theta_l, r_e}) + \lambda\theta_e$, *where* $\mu_e(\pi_{\theta_l, \theta_e}) \triangleq E^{\pi_{\theta_l, \theta_e}}[\sum_{t=0}^{T-1} \gamma^t \nabla_{\theta_e} r_{\theta_e}(s^t, a^t)]$, $\hat{\mu}_e(D_{\theta_l, r_e}) \triangleq \frac{1}{d}\sum_{i=1}^{d}\sum_{t=0}^{T-1} \gamma^t \nabla_{\theta_e} r_{\theta_e}(s^{it}, a^{it})$, *and* $(s^{it}, a^{it}) \in \zeta_i \in D_{\theta_l, r_e}$

The inner loop uses $t_k$-step projected gradient descent $\theta_e(t+1) = \Pi_\Theta(\theta_e(t) - \beta_t \nabla_{\theta_e} L(\theta_l(k), \theta_e(t)))$ to obtain an expert reward estimate $r_{\theta_e(t_k-1)}$.

### 5.2 OUTER LOOP

At $k$-th iteration of the outer loop, the learner updates the parameter $\theta_l(k)$ via projected SGD. Similar to the inner loop, this update requires computing the hypergradient $\nabla f(\theta_l(k), \theta_e(k))$. The analytical form of $\nabla f(\theta_l(k), \theta_e(k))$, which is widely used in bi-level optimization problems, is presented in the following lemma.

**Lemma 2.** *The analytical expression of the hypergradient* $\nabla f(\theta_l, \theta_e)$ *used for updating* $\theta_l$ *is* $\nabla_{\theta_l} f(\theta_l, \theta_e) - \nabla_{\theta_l \theta_e}^2 L(\theta_l, \theta_e)[\nabla_{\theta_e}^2 L(\theta_l, \theta_e)]^{-1}\nabla_{\theta_e} f(\theta_l, \theta_e)$ *(Xu & Zhu, 2023; Ghadimi & Wang, 2018; Colson et al., 2007).*

---

**Algorithm 1** General Scenario Interactive Inverse Reinforcement Learning (GSIIRL)

---

Initializes $\theta_l(0) \in \Theta, \theta_e(0) \in \Theta$, step size sequence $\{\alpha_k\}, \{\beta_t\}$, regularization parameter $\lambda$ and integer sequence $\{t_k\}$
**for** $k = 0, 1, \cdots, K-1$ **do**
    $\theta_e(0) = \theta_e(k)$
    Samples the trajectory set $D_{\theta_l(k), r_e}$
    **for** $t = 0, \cdots, t_k - 1$ **do**
        Calculates $\nabla_{\theta_e} L(\theta_l(k), \theta_e(t))$ following Lemma 1
        $\theta_e(t+1) = \Pi_\Theta(\theta_e(t) - \beta_t \nabla_{\theta_e} L(\theta_l(k), \theta_e(t)))$
    **end for**
    $\theta_e(k) = \theta_e(t_k - 1)$
    Initializes the random vector $\Delta(k)$ and the positive scalar $p(k)$
    Gets $\hat{\nabla}^2_{\theta_l \theta_e} L(\theta_l(k), \theta_e(k))$, $\hat{\nabla}^2_{\theta_e} L(\theta_l(k), \theta_e(k))$, $\hat{\nabla}_{\theta_l} f(\theta_l, \theta_e)$ and $\hat{\nabla}_{\theta_e} f(\theta_l, \theta_e)$ through SPSA approximation following equation (4)
    Calculates $[\hat{\nabla}^2_{\theta_e} L(\theta_l(k), \theta_e(k))]^{-1} \hat{\nabla}_{\theta_e} f(\theta_l(k), \theta_e(k))$ following equation (5)
    Calculates $\hat{\nabla} f(\theta_l(k), \theta_e(k))$ following equation (6)
    $\theta_l(k+1) = \Pi_\Theta(\theta_l(k) - \alpha_k \hat{\nabla} f(\theta_l(k), \theta_e(k)))$
**end for**

---

Computing $\nabla f(\theta_l(k), \theta_e(k))$ involves the partial gradients $\nabla_{\theta_l} f(\theta_l, \theta_e), \nabla_{\theta_e} f(\theta_l, \theta_e)$, the Jacobian $\nabla^2_{\theta_l \theta_e} L(\theta_l, \theta_e)$ and the inverse Hessian $[\nabla^2_{\theta_e} L(\theta_l, \theta_e)]^{-1}$. However, directly calculating these quantities has a large computational complexity of $\mathcal{O}(T^2)$. To address this issue, we adopt the SPSA method, which reduces computational complexity to $\mathcal{O}(T)$. A detailed discussion on the computational complexity reduction is provided later in Theorem 1. Using SPSA, the learner obtains the estimated hypergradient $\hat{\nabla} f(\theta_l(k), \theta_e(k))$, which serves as the gradient approximation in the projected SGD update. To calculate $\hat{\nabla} f(\theta_l(k), \theta_e(k))$, the learner estimates $\hat{\nabla}_{\theta_l} f(\theta_l(k), \theta_e(k))$, $\hat{\nabla}^2_{\theta_l \theta_e} L(\theta_l(k), \theta_e(k))$, $\hat{\nabla}^2_{\theta_e} L(\theta_l(k), \theta_e(k))$, and $\hat{\nabla}_{\theta_e} f(\theta_l(k), \theta_e(k))$. Let us take $\hat{\nabla}^2_{\theta_e} L(\theta_l(k), \theta_e(k))$ as an example to illustrate SPSA. According to the equation (2.2) in (Spall, 1992), it is approximated as follows:

$$\hat{\nabla}^2_{\theta_e} L(\theta_l(k), \theta_e(k)) = \begin{bmatrix} \frac{\Delta_{d_k}}{2p\Delta_1(k)} & \cdots & \frac{\Delta_{d_k}}{2p\Delta_m(k)} \end{bmatrix}^T, \tag{4}$$

where $\Delta_{d_k} = \nabla_{\theta_e} L(\theta_l(k), \theta_e(k) + p(k)\Delta(k)) - \nabla_{\theta_e} L(\theta_l(k), \theta_e(k) - p(k)\Delta(k))$, the perturbation $\Delta(k) \in \mathbb{R}^m$ is a vector of $m$ mutually independent zero-mean random variables and each element of $\Delta(k)$ satisfies $|\Delta_i(k)| \leq \alpha_0$, $E|\Delta_i^{-1}(k)| \leq \alpha_1$, $i = 1, \cdots, m$ with $\alpha_0, \alpha_1$ as positive constants. The parameter $p(k)$ is a positive scalar.

Referring to equation (4), the computation of $\hat{\nabla}_{\theta_l} f(\theta_l(k), \theta_e(k))$ requires $f(\theta_l(k), \theta_e(k))$ with weight perturbations on $\theta_l(k)$. Analogously, computing $\hat{\nabla}_{\theta_e} f(\theta_l(k), \theta_e(k))$, $\hat{\nabla}^2_{\theta_e} L(\theta_l(k), \theta_e(k))$, and $\hat{\nabla}^2_{\theta_l \theta_e} L(\theta_l(k), \theta_e(k))$ requires $f(\theta_l(k), \theta_e(k))$, $\nabla_{\theta_e} L(\theta_l(k), \theta_e(k))$, and $\nabla_{\theta_l} L(\theta_l(k), \theta_e(k))$ with weight perturbations on $\theta_e(k)$, respectively. Notice that the analytical expression of $f(\theta_l(k), \theta_e(k))$ and $\nabla_{\theta_e} L(\theta_l(k), \theta_e(k))$ are shown in optimization problem (2) and Lemma 1 respectively. Similarly to Lemma 1, the gradient $\nabla_{\theta_l} L(\theta_l(k), \theta_e(k)) \triangleq \mu_l(\pi_{\theta_l(k), \theta_e(k)}) - \hat{\mu}_l(D_{\theta_l(k), r_e})$ and the proof is given in the Appendix A.4. The expectation $\mu_l(\pi_{\theta_l, \theta_e}) \triangleq E^{\pi_{\theta_l, \theta_e}}[\sum_{t=0}^{T-1} \gamma^t \nabla_{\theta_l} r_{\theta_l}(s^t, a^t)]$ is the reward gradient expectation of the learner and the empirical expectation $\hat{\mu}_l(D_{\theta_l, r_e}) \triangleq \frac{1}{d} \sum_{i=1}^d \sum_{t=0}^{T-1} \gamma^t \nabla_{\theta_l} r_{\theta_l}(s^{it}, a^{it}), (s^{it}, a^{it}) \in \zeta_i \in D_{\theta_l, r_e}$ is the estimated reward gradient expectation of the learner.

To compute the hypergradient $\hat{\nabla} f(\theta_l(k), \theta_e(k))$, we must ensure that the estimated Hessian $\hat{\nabla}^2_{\theta_e} L(\theta_l(k), \theta_e(k))$ is invertible. Since the negative log-likelihood function is convex and the $L_2$ regularization term is strongly convex, the overall function $L(\theta_l(k), \theta_e(k))$ is strongly convex. Hence its Hessian is positive definite. By choosing suitable $\Delta(k)$ and $p(k)$, we ensure that the estimated Hessian $\hat{\nabla}^2_{\theta_e} L(\theta_l(k), \theta_e(k))$ is positive definite. Since directly inverting a matrix is com-

putationally expensive, we use the conjugate gradient method to compute the product

$$
[\hat{\nabla}^2_{\theta_e} L(\theta_l(k), \theta_e(k))]^{-1} \hat{\nabla}_{\theta_e} f(\theta_l(k), \theta_e(k))
$$
$$
= \min_u \frac{1}{2} u^T \hat{\nabla}^2_{\theta_e} L(\theta_l(k), \theta_e(k)) u - u^T \hat{\nabla}_{\theta_e} f(\theta_l(k), \theta_e(k)). \tag{5}
$$

Finally, using $\hat{\nabla}_{\theta_l} f(\theta_l(k), \theta_e(k))$, $\hat{\nabla}^2_{\theta_l \theta_e} L(\theta_l(k), \theta_e(k))$, and the result from conjugate gradient, we compute the estimated hypergradient

$$
\hat{\nabla} f(\theta_l(k), \theta_e(k))
$$
$$
= \hat{\nabla}_{\theta_l} f(\theta_l(k), \theta_e(k)) - \hat{\nabla}^2_{\theta_l \theta_e} L(\theta_l(k), \theta_e(k)) [\hat{\nabla}^2_{\theta_e} L(\theta_l(k), \theta_e(k))]^{-1} \hat{\nabla}_{\theta_e} f(\theta_l(k), \theta_e(k)). \tag{6}
$$

After $K$ iterations, the outer loop terminates, yielding $r_{\theta_l(K)}$, $r_{\theta_e(K)}$ and $\pi_{\theta_l(K), \theta_e(K)}$.

## 6 ANALYTICAL RESULT

In this section, we present our analytical results regarding computational complexity reduction and convergence rate. To facilitate our analysis, we impose the following assumption on the estimated reward functions $r_{\theta_l}$ and $r_{\theta_e}$:

**Assumption 1.** *The estimated expert-dependent reward function $r_{\theta_l}$ and the estimated expert reward function $r_{\theta_e}$ are fourth differentiable, i.e., $C^4$.*

Since $\theta_l$ and $\theta_e$ lie in compact sets, Assumption 1 implies that the derivatives of the reward functions $r_{\theta_l}$ and $r_{\theta_e}$ are bounded. Such boundedness of higher-order derivatives is a standard assumption in the literature and has been widely adopted in bi-level optimization (Jin et al., 2020; Zeng et al., 2022; Liu & Zhu, 2023a), RL (Wang et al., 2019; Zhang et al., 2020), and IRL (Liu & Zhu, 2023b).

### 6.1 COMPUTATIONAL COMPLEXITY

The computation complexities of computing $\nabla f(\theta_l, \theta_e)$ and $\hat{\nabla} f(\theta_l, \theta_e)$ are shown in Theorem 1.

**Theorem 1.** *Consider $T$ as the decision factor, the computational complexity of computing $\nabla f(\theta_l, \theta_e)$ is $\mathcal{O}(T^2)$ and that of computing $\hat{\nabla} f(\theta_l, \theta_e)$ is $\mathcal{O}(T)$.*

Applying SPSA to approximate $\nabla f(\theta_l, \theta_e)$ reduces the per-iteration computational complexity of the upper-level problem from $\mathcal{O}(T^2)$ to $\mathcal{O}(T)$. The proof of Theorem 1 is in the Appendix A.8. Compared to finite-difference methods, SPSA requires fewer policies for the approximation. In SPSA, a single random perturbation of $\theta_l$ or $\theta_e$ yields two policies (one for the positive perturbation and one for the negative perturbation). In contrast, the finite difference approach requires two policies for each parameter dimension. Each policy is generated via MARL, so the computational cost of producing these policies is non-negligible. As SPSA requires fewer policies, its overall computational cost of SPSA is lower than that of finite difference methods. Finite-difference methods, in turn, have a lower computational cost than explicitly computing gradients.

### 6.2 CONVERGENCE RATE

The convergence of our algorithm is shown in Theorem 2.

**Theorem 2.** *Suppose Assumption 1 holds, by choosing $p(k) = \frac{1}{k}$, $\alpha_k = \frac{1}{L_f \sqrt{K}}$, $t_k = \lceil \frac{\sqrt[4]{k+1}}{2} \rceil$, the following convergence guarantee holds: $\frac{1}{K} \sum_{k=0}^{K-1} E[\|\nabla f(\theta_l(k), \theta_e^*(\theta_l(k)))\|^2] \leq \mathcal{O}(\frac{1}{\sqrt{K}})$.*

Theorem 2 indicates that the expected hypergradient decays at a rate of $\mathcal{O}(\frac{1}{\sqrt{K}})$, matching the convergence rate of standard bilevel optimization (Ghadimi & Wang, 2018). This implies that the bias introduced by SPSA does not slow convergence compared to standard bilevel optimization. Corollary 2.1 shows that the linear reward function $r_{\theta_e}$ is a sufficient condition for the convergence of the policy $\pi_{\theta_e}$. Convergence of the cumulative reward difference has been widely used to infer convergence of the learned expert policy (Rhinehart & Kitani, 2017; Renard et al., 2024).

**Corollary 2.1.** *If the reward function $r_{\theta_e}$ is linear, the cumulative reward difference between the learned expert policy $\pi_{\theta_e}$ and $\pi_e$ decreases at a rate $\mathcal{O}(\frac{1}{\sqrt[4]{K}})$.*

# 7 EXPERIMENT

In our experiments, we evaluate GSIIRL on four widely used environments. The Multi-Agent Particle Environment (MPE) (Lowe et al., 2017; Terry et al., 2021; Yu et al., 2022) involves particle agents that can move, communicate, observe one another, push each other, and interact with fixed landmarks. The StarCraft Multi-Agent Challenge (SMAC) (Samvelyan et al., 2019; Yu et al., 2022; Rashid et al., 2020) is designed based on the real-time strategy game StarCraft II. The Human-Robot Interaction environment (El-Shamouty et al., 2020; Fan et al., 2020; Zhu & Zhang, 2021) models a navigation scenario in which a robot interacts with a human; the corresponding results are reported in Appendix Section A.9.3. The security environment (Zhang et al., 2019; 2021) involves automated defense systems; the results are presented in Appendix Section A.9.4.

We compare GSIIRL with four baseline algorithms: the **MARL** algorithm (Lowe et al., 2017), the **MA-IRL** algorithm (Ziebart et al., 2010), the cooperative interactive IRL (**CIRL**) algorithm (Büning et al., 2022), and the maximum likelihood IRL (**ML-IRL**) algorithm (Zeng et al., 2022). Since none of these baseline algorithms can directly address the interactive IRL problem, we adapt the experimental setup to make them applicable. Specifically, for MARL, we assume that both the learner and the expert know their reward functions. For MA-IRL, we provide demonstrations sampled according to ground-truth reward functions of both the learner and the expert. For CIRL, we assume that the expert-dependent reward function of the learner is identical to the expert's reward function. For ML-IRL, we assume that the learner has access to demonstrations generated by a policy through MARL with the learner's initial and the ground-truth expert reward functions.

The MARL baseline serves to illustrate the best achievable performance among all related algorithms. The MA-IRL baseline evaluates whether our algorithm can reach similar performance with less information. The CIRL baseline demonstrates that cooperative interactive IRL algorithms are only suitable for fully cooperative cases of interactive IRL problems. The ML-IRL baseline shows that conventional IRL algorithms are not well-suited for interactive IRL problems.

## 7.1 MULTI-AGENT PARTICLE ENVIRONMENTS

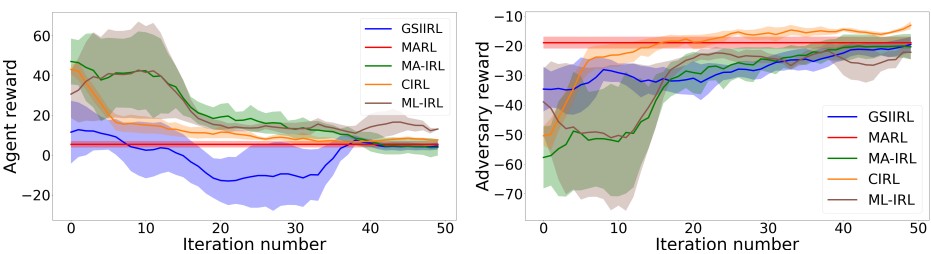

Figure 2: MPE simulation results. **Left**: Agent's reward. **Right**: Adversary's reward. The horizontal lines show the cumulative rewards from a MARL method that has access to the ground-truth reward functions after convergence. In contrast, other methods learn the reward functions from demonstrations and interactions, respectively. In each iteration, we compute the policies of both the adversary and the agents via MARL using the currently learned reward functions. We then evaluate these policies using the ground-truth reward functions.

We use the physical deception environment in the MPE for our simulation. This environment includes one adversary, two cooperating agents, and two landmarks. One of the landmarks is the target landmark. The cooperating agents know which landmark is the target, whereas the adversary does not. The cooperating agents aim to have one of them approach the target as closely as possible while preventing the adversary from reaching it. Conversely, the adversary aims to identify and reach the target landmark. We treat the two cooperating agents as a single learner and the adversary as the expert. Both the learner and the expert have continuous state and action spaces. The learner's state space is 10-dimensional and the expert's is 8-dimensional, and each has a 5-dimensional action space. Other details are provided in Appendix Section A.9.1.

Because the two agents cooperate and share a common reward, we plot a single cumulative reward curve for both. Figure 2 presents results for two randomly initialized reward functions updated over 50 iterations. Initially, both the adversary and the agents act randomly, and the cumulative

rewards differ significantly from those of the MARL baseline. As more data are collected, the agents gradually learn their own goal and the adversary's goal, leading the cumulative rewards of GSIIRL and MA-IRL to converge to that of MARL. In contrast, CIRL and ML-IRL fail to do so.

## 7.2 STARCRAFT MULTI-AGENT CHALLENGE

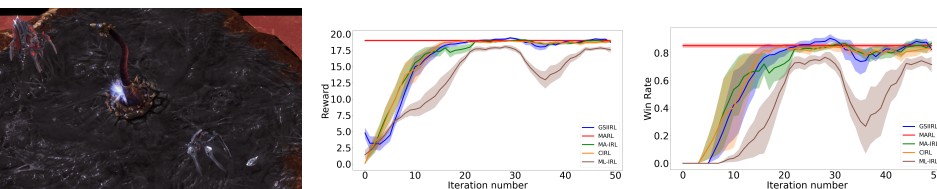

Figure 3: SMAC simulation results. **Left**: A screenshot of the game. **Middle**: Cumulative reward. **Right**: Win rate. The cumulative rewards are computed in the same manner as described in Figure 2. Similarly, the win rates are computed based on the policies corresponding to the learned reward functions. The agents are considered victorious when they defeat the enemy.

We use the scenario 2s_vs_1sc from the SMAC for our experiment. In this scenario, two agents co-operate to defeat a single, more powerful enemy unit (which is controlled by a built-in AI). Because the environment does not expose the enemy's state or action information, the enemy's reward function cannot be learned directly. We therefore designate one agent as the expert (with access to the ground-truth reward function and the optimal cooperative strategy to defeat the enemy) and the other agent as the learner. The learner's goal is to infer the expert's strategy and learn to cooperate with the expert accordingly. Since this setting is fully cooperative, the agents share a common reward; accordingly, we ignore any expert-independent component of the learner's reward and consider only the expert-dependent component. Both agents have a continuous 17-dimensional state space and a discrete action space of size 7. Additional implementation details are provided in Appendix A.9.2.

Because the two agents share the same reward, we report a single cumulative reward curve. The middle panel of Figure 3 shows that the cumulative rewards of GSIIRL and MA-IRL converge to that achieved by MARL, mirroring the trend observed in Figure 2. Since this scenario is fully cooperative, the cumulative reward of CIRL also converges to that of the MARL, whereas ML-IRL does not. We further evaluate performance using the agents' win rate. At each iteration, we obtain the agents' policies via MARL with the current learned reward functions. Each learned policy is then evaluated over 100 independent trials, and the win rate (the fraction of trials in which the agents defeat the enemy) is recorded. The right panel of Figure 3 shows that the win rates of GSIIRL, MA-IRL, and CIRL converge to that of the MARL baseline, while ML-IRL fails to do so.

## 7.3 RESULTS ANALYSIS

From Figures 2 and 3, we conclude that CIRL is effective only in fully cooperative cases, and ML-IRL is not suitable for interactive IRL problems. We therefore focus on comparing the cumulative rewards achieved by MA-IRL, GSIIRL, and MARL across the four experiments. Figure 2 and Figure 3 show that the cumulative rewards of MA-IRL and GSIIRL converge to those of MARL. The final cumulative rewards of GSIIRL are comparable to those of MA-IRL and MARL, differing by at most 5%, and a table with numerical cumulative reward comparison in shown in Appendix A.9.5. Notably, GSIIRL requires less information than MA-IRL: it relies only on the expert's interactions with the environment and the corresponding values received during the interactions. In contrast, MA-IRL requires demonstrations generated using the ground-truth reward functions of both the learner and the expert, whereas MARL assumes full knowledge of the ground-truth reward functions.

## 8 CONCLUSION

We develop a general interactive IRL framework that enables a learner to infer an expert's reward function and learn an appropriate interaction policy through interactions with the expert. This frame-work releases the strong assumption on the learner's and the expert's reward functions. We propose the GSIIRL algorithm and provide a theoretical analysis of its convergence rate. Experiments in both continuous and discrete environments demonstrate the effectiveness of GSIIRL.

## 9    Ethics statement

This work does not involve human subjects or sensitive personal data. All experiments are conducted on publicly available simulation benchmarks.

## 10    Reproducibility statement

The details of the experimental setup are provided in Appendix A.9, and the source code is included in the supplementary material to facilitate reproducibility.

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

# A APPENDIX

## A.1 NOTION AND NOTATIONS

Define $f(\theta_l, \theta_e) \triangleq J_{ed}(\pi_{\theta_l, \theta_e}) + J_{ei}(\pi_{\theta_l, \theta_e})$, where $J_{ed}(\pi_{\theta_l, \theta_e}) \triangleq E^{\pi_{\theta_l, \theta_e}}[\sum_{t=0}^{T-1} \gamma^t r_{\theta_l}(s^t, a^t)]$ is the cumulative expert-dependent reward of the learner and $J_{ei}(\pi_{\theta_l, \theta_e}) \triangleq E^{\pi_{\theta_l, \theta_e}}[\sum_{t=0}^{T-1} \gamma^t r_{li}(s_l^t, a_l^t)]$ is the cumulative expert-independent reward of the learner. Define the reward gradient expectation of the expert from the state-action pair $(s, a)$ as $\mu_e(s, a) \triangleq E^{\pi_{\theta_l, \theta_e}}[\sum_{t=0}^{T-1} \gamma^t \nabla_{\theta_e} r_e^{\theta_e}(s^t, a^t)|s^0 = s, a^0 = a]$, the reward gradient expectation of the expert from the state $s$ as $\mu_e(s) \triangleq \int_{a_l \in a_l} \int_{a_e \in a_e} \mu_e(s, a) da_l da_e, \mu_e(s, a) \triangleq E^{\pi_{\theta_l, \theta_e}}[\sum_{t=0}^{T-1} \gamma^t \nabla_{\theta_e} r_e^{\theta_e}(s^t, a^t)|s^0 = s]$. Analogously, define the reward gradient expectation of the learner as $\mu_l(\pi_{\theta_l, \theta_e}) \triangleq E^{\pi_{\theta_l, \theta_e}}[\sum_{t=0}^{T-1} \gamma^t \nabla_{\theta_l} r_{\theta_l}(s^t, a^t)]$, the reward gradient expectation of the learner from the state-action pair $(s, a)$ as $\mu_l(s) \triangleq E^{\pi_{\theta_l, \theta_e}}[\sum_{t=0}^{T-1} \gamma^t \nabla_{\theta_l} r_{\theta_l}(s^t, a^t)|s^0 = s]$, $\mu_l(s, a) \triangleq E^{\pi_{\theta_l, \theta_e}}[\sum_{t=0}^{T-1} \gamma^t \nabla_{\theta_l} r_{\theta_l}(s^t, a^t)|s^0 = s, a^0 = a]$, the reward gradient expectation of the learner from the state $s$ as $\mu_l(s) \triangleq \int_{a_l \in a_l} \int_{a_e \in a_e} \mu_l(s, a) da_l da_e$. Define the cumulative expert-independent reward of the learner from the state-action pair $(s, a)$ as $J_{ei}(s, a) \triangleq E^{\pi_{\theta_l, \theta_e}}[\sum_{t=0}^{T-1} \gamma^t r_{li}(s_l^t, a_l^t)|s^0 = s, a^0 = a]$ and define the cumulative expert-independent reward of the learner from the state $s$ as $J_{ei}(s) \triangleq E^{\pi_{\theta_l, \theta_e}}[\sum_{t=0}^{T-1} \gamma^t r_{li}(s_l^t, a_l^t)|s^0 = s]$. Define $P_0(s)$ as the probability distribution of the initial state.

During the proofs, symbols $(i)$ to $(viii)$ are used to represent what theorems or methods are used to get the current step. The symbol $(i)$ represents chain rule, the symbol $(ii)$ represents the linearity of expectation, the symbol $(iii)$ represents the close form of geometric series, the symbol $(iv)$ represents the triangle inequality, the symbol $(v)$ represents the hölder's inequality, the symbol $(vi)$ represents Taylor's theorem, the symbol $(vii)$ means the usage of other equations in this paper, and the symbol $(viii)$ means keeping expansion.

## A.2 FUNDAMENTAL RESULT FOR POLICY

This section lists the expressions for important gradients in continuous state-action space. Based on the idea of the soft Q learning (Haarnoja et al., 2017), we can get:

$$Q_{\theta_l, \theta_e}^{soft}(s, a) = r_{\theta_l}(s, a) + r_{li}(s, a) + r_e^{\theta_e}(s, a) + \gamma \int_{s' \in S} P(s'|s, a) V^{soft}(s') ds', \quad (7)$$

$$V_{\theta_l, \theta_e}^{soft}(s) = \ln \int_{a_l \in a_l} \int_{a_e \in a_e} \exp(Q^{soft}(s, a)) da_l da_e, \quad (8)$$

$$\pi_{\theta_l, \theta_e}(a_l, a_e|s) = \frac{\exp(Q^{soft}(s, a))}{\exp(V^{soft}(s))}. \quad (9)$$

As the lower-level optimization problem is a maximum likelihood problem, it is important to know $\nabla_{\theta_e} \ln(\pi_{\theta_l, \theta_e})$ for the SGD of the lower-level optimization problem. The process for getting $\nabla_{\theta_e} \ln(\pi_{\theta_l, \theta_e})$ is shown below.

From the expression of $\pi_{\theta_l, \theta_e}$, we can get $\nabla_{\theta_e} \ln(\pi_{\theta_l, \theta_e}) = \nabla_{\theta_e} Q^{soft}(s, a) - \nabla_{\theta_e} V^{soft}(s)$, therefore, we can calculate $\nabla_{\theta_e} Q^{soft}(s, a)$ and $\nabla_{\theta_e} V^{soft}(s)$ separately.

Based on the equation of $V^{soft}(s)$, the gradient $\nabla_{\theta_e} V^{soft}(s)$ could be calculated as follows:

$$\nabla_{\theta_e} V^{soft}(s),$$

$$\stackrel{(i)}{=} \frac{\int_{a_l \in a_l} \int_{a_e \in a_e} \nabla_{\theta_e} \exp(Q^{soft}(s,a)) da_l da_e}{\int_{a_l \in a_l} \int_{a_e \in a_e} \exp(Q^{soft}(s,a)) da_l da_e},$$

$$\stackrel{(vii)}{=} \frac{\int_{a_l \in a_l} \int_{a_e \in a_e} \exp(Q^{soft}(s,a)) \nabla_{\theta_e} Q^{soft}(s,a) da_l da_e}{\exp(V^{soft}(s))},$$

$$\stackrel{(vii)}{=} \int_{a_l \in a_l} \int_{a_e \in a_e} \pi_{\theta_l,\theta_e}(a_l,a_e|s)(\nabla_{\theta_e} r_e(s,a)$$

$$+ \gamma \int_{s' \in S} P(s'|s,a) \nabla_{\theta_e} V^{soft}(s')ds')da_l da_e,$$

$$\stackrel{(viii)}{=} \int_{a_l \in a_l} \int_{a_e \in a_e} \pi_{\theta_l,\theta_e}(a_l,a_e|s)(\nabla_{\theta_e} r_e(s,a) + \gamma \int_{s' \in S} P(s'|s,a)$$

$$(\int_{a'_l \in a_l} \int_{a'_e \in a_e} \pi_{\theta_l,\theta_e}(a'_l,a'_e|s')[\nabla_{\theta_e} r_e(s',a'_l,a'_e)$$

$$+ \gamma \int_{s'' \in S} P(s''|s',a'_l,a'_e) \nabla_{\theta_e} V^{soft}(s'')ds'']da'_l da'_e ds')da_l da_e,$$

$$= E^{\pi_{\theta_l,\theta_e}}[\sum_{t=0}^{T-1} \gamma^t \nabla_{\theta_e} r_e(s^t,a^t)|s^0 = s],$$

$$= \mu_e(s), \tag{10}$$

where the first $(vii)$ uses the expression in equation (8) and (9), and the second $(iii)$ uses the expression in equation (7).
Similarly, we can get $\nabla_{\theta_e} Q^{soft}(s,a)$ from $Q^{soft}(s,a)$.

$$\nabla_{\theta_e} Q^{soft}(s,a),$$

$$\stackrel{(vii)}{=} \nabla_{\theta_e} r_e(s,a) + \gamma \int_{s' \in S} P(s'|s,a) \nabla_{\theta_e} V^{soft}(s')ds')da_l da_e,$$

$$\stackrel{(viii)}{=} E^{\pi_{\theta_l,\theta_e}}[\sum_{t=0}^{T-1} \gamma^t \nabla_{\theta_e} r_e(s^t,a^t)|s^0 = s, a^0 = a],$$

$$= \mu_e(s,a), \tag{11}$$

where $(vii)$ uses the expression in equation (7).
By summing the results of equation (10) and (11), we can get $\nabla_{\theta_e} \ln(\pi_{\theta_l,\theta_e})$ as follows:

$$\nabla_{\theta_e} \ln(\pi_{\theta_l,\theta_e}(a_l,a_e|s)) = \nabla_{\theta_e} Q^{soft}(s,a) - \nabla_{\theta_e} V^{soft}(s) = \mu_e(s,a) - \mu_e(s). \tag{12}$$

The way to get $\nabla_{\theta_l} \ln(\pi_{\theta_l,\theta_e})$ is same as that of $\nabla_{\theta_e} \ln(\pi_{\theta_l,\theta_e})$, and the gradient $\nabla_{\theta_l} \ln(\pi_{\theta_l,\theta_e}(a_l,a_e|s)) = \mu_l(s,a) - \mu_l(s)$.

### A.3 THE PROOF OF LEMMA 1

In this section, we derived gradients that are necessary for our method.

Define $P_D(s^t = s, a_l^t = a_l, a_e^t = a_e) \triangleq \begin{cases} 1 & s^t = s, a_l^t = a_l, a_e^t = a_e \\ 0 & \text{otherwise} \end{cases}$ , where $(s,a) \in D_{\theta_l,r_e}$, as the probability of $(s,a)$ occurring at time $t$ in the demonstration set $D_{\theta_l,r_e}$. With the fundamental

result of the policy, we can derive the $\nabla_{\theta_e} L(\theta_l, \theta_e)$ and prove the Lemma 1 as follows:

$$\nabla_{\theta_e} L(\theta_l, \theta_e),$$

$$= -\sum_{t=0}^{T-1} \gamma^t \int_{s \in S} \int_{a_l \in a_l} \int_{a_e \in a_e} P_D(s^t = s, a_l^t = a_l, a_e^t = a_e) \nabla_{\theta_e} \ln(\pi_{\theta_l, \theta_e}) da_e da_l ds + \lambda \theta_e,$$

$$\overset{(vii)}{=} -\sum_{t=0}^{T-1} \gamma^t \int_{s \in S} \int_{a_l \in a_l} \int_{a_e \in a_e} P_D(s^t = s, a_l^t = a_l, a_e^t = a_e)(\nabla_{\theta_e} r_e(s, a)$$

$$+ E^{\pi_{\theta_l, \theta_e}} \Big[ \sum_{t'=t+1}^{T-1} \gamma^{t'-t-1} \nabla_{\theta_e} r_e(s^t, a^t) | s^t = s, a_l^t = a_l, a_e^t = a_e \Big]$$

$$- E^{\pi_{\theta_l, \theta_e}} \Big[ \sum_{t'=t}^{T-1} \gamma^{t'-t} \nabla_{\theta_e} r_e(s^t, a^t) | s^t = s \Big]) da_e da_l ds + \lambda \theta_e,$$

$$\overset{(ii)}{=} -\sum_{t=0}^{T-1} \gamma^t \int_{s \in S} \int_{a_l \in a_l} \int_{a_e \in a_e} P_D(s^t = s, a_l^t = a_l, a_e^t = a_e)(\nabla_{\theta_e} r_e(s, a) da_e da_l ds$$

$$- \int_{s \in S} \int_{a_l \in a_l} \int_{a_e \in a_e} P_D(s^0 = s, a_l^0 = a_l, a_e^0 = a_e)$$

$$E^{\pi_{\theta_l, \theta_e}} \Big[ \sum_{t=0}^{T-1} \gamma^t \nabla_{\theta_e} r_e(s^t, a^t) | s^0 = s \Big]) da_e da_l ds + \lambda \theta_e,$$

$$= \mu_e(\pi_{\theta_l, \theta_e}) - \hat{\mu}_e(D_{\theta_l, r_e}) + \lambda \theta_e,$$

where $(vii)$ uses the expression of equation (12).

### A.4 OTHER NEEDED GRADIENTS

**Lemma 3.** *Suppose Assumption 1 holds, the first-order gradients* $\nabla_{\theta_l} L(\theta_l, \theta_e) = \mu_l(\pi_{\theta_l, \theta_e}) - \hat{\mu}_l(D_{\theta_l, r_e}), \nabla_{\theta_l} f(\theta_l, \theta_e) = E^{\pi_{\theta_l, \theta_e}} [\sum_{t=0}^{T-1} \gamma^t [(\mu_l(s^t, a^t) - \mu_l(s^t))(J_{ed}(s^t, a^t) + J_{ei}(s^t, a^t))^T]] + \mu_l(\pi_{\theta_l, \theta_e}), \nabla_{\theta_e} f(\theta_l, \theta_e) = E^{\pi_{\theta_l, \theta_e}} [\sum_{t=0}^{T-1} \gamma^t [(\mu_e(s^t, a^t) - \mu_e(s^t))(J_{ed}(s^t, a^t) + J_{ei}(s^t, a^t))^T]].$ *The second-order gradients* $\nabla^2_{\theta_l \theta_e} L(\theta_l, \theta_e) = E^{\pi_{\theta_l, \theta_e}} [\sum_{t=0}^{T-1} \gamma^t (\mu_e(s, a) - \mu_e(s)) \mu_l(s, a)^T],$ $\nabla^2_{\theta_e \theta_e} L(\theta_l, \theta_e) = E^{\pi_{\theta_l, \theta_e}} [\sum_{t=0}^{T-1} \gamma^t [(\mu_e(s^t, a^t) - \mu_e(s^t)) \mu_e(s^t, a^t)^T + \nabla^2_{\theta_e} r_e^{\theta_e}(s^t, a^t)]] - \frac{1}{d} \sum_{i=0}^{d} \sum_{t=0}^{T-1} \gamma^t \nabla_{\theta_e} r_e^{\theta_e}(s^{it}, a^{it}) + \lambda$

*Proof.* Through the same process of calculating $\nabla_{\theta_e} L(\theta_l, \theta_e)$, we can get the gradient $\nabla_{\theta_l} L(\theta_l, \theta_e)$ following the same process in Lemma 1, where $\mu_l(\pi_{\theta_l, \theta_e}) \triangleq E^{\pi_{\theta_l, \theta_e}} [\sum_{t=0}^{T-1} \gamma^t \nabla_{\theta_l} r_{ld}(s^t, a^t)].$

The process for getting $\nabla_{\theta_l} f(\theta_l, \theta_e)$ is shown below.
From the equation of $f(\theta_l, \theta_e)$, we can see that $\nabla_{\theta_l} f(\theta_l, \theta_e) = \nabla_{\theta_l} J_{ed}(\pi_{\theta_l, \theta_e}) + \nabla_{\theta_l} J_{ei}(\pi_{\theta_l, \theta_e}),$ then $\nabla_{\theta_l} J_{ed}(\pi_{\theta_l, \theta_e})$ and $\nabla_{\theta_l} J_{ei}(\pi_{\theta_l, \theta_e})$ can be calculated separately.

The calculation of deriving $\nabla_{\theta_l} J_{ed}(\pi_{\theta_l, \theta_e})$ is as follows:

$\nabla_{\theta_l} J_{ed}(\pi_{\theta_l, \theta_e})$,

$$\overset{(i)}{=} \int_{s^0 \in S} P_0(s^0) \int_{a_l^0 \in a_l} \int_{a_e^0 \in a_e} [\nabla_{\theta_l} \pi(a_l^0, a_e^0 | s^0) J_{ed}(s^0, a_l^0, a_e^0)^T$$

$$+ \pi(a_l^0, a_e^0 | s^0) \nabla_{\theta_l} J_{ed}(s^0, a_l^0, a_e^0)^T] da_l^0 da_e^0 ds^0,$$

$$= \int_{s^0 \in S} P_0(s^0) \int_{a_l^0 \in a_l} \int_{a_e^0 \in a_e} [\nabla_{\theta_l} \pi(a_l^0, a_e^0 | s^0) J_{ed}(s^0, a_l^0, a_e^0)^T + \pi(a_l^0, a_e^0 | s^0)(\nabla_{\theta_l} r_{\theta_l}(s^0, a_l^0, a_e^0)$$

$$+ \gamma \int_{s^1 \in S} P(s^1 | s^0, a_l^0, a_e^0) \nabla_{\theta_l} J_{ed}(s^1)) ds^1] da_l^0 da_e^0 ds^0,$$

$$\overset{(viii)}{=} \int_{s^0 \in S} P_0(s^0) \int_{a_l^0 \in a_l} \int_{a_e^0 \in a_e} \{\nabla_{\theta_l} \pi(a_l^0, a_e^0 | s^0) J_{ed}(s^0, a_l^0, a_e^0)^T + \pi(a_l^0, a_e^0 | s^0)[\nabla_{\theta_l} r_{\theta_l}(s^0, a_l^0, a_e^0)$$

$$+ \gamma \int_{s^1 \in S} P(s^1 | s^0, a_l^0, a_e^0) \int_{a_l^1 \in a_l} \int_{a_e^1 \in a_e} \nabla_{\theta_l} \pi(a_l^1, a_e^1 | s^1) J_{ed}(s^1, a_l^1, a_e^1)^T$$

$$+ \pi(a_l^1, a_e^1 | s^1)(\nabla_{\theta_l} r_{\theta_l}(s^1, a_l^1, a_e^1) + \gamma \int_{s^2 \in S} P(s^2 | s^1, a_l^1, a_e^1) \nabla_{\theta_l} J_{ed}(s^2)) ds^2] ds^1\} da_l^0 da_e^0 ds^0,$$

$$= E^{\pi_{\theta_l, \theta_e}} [\sum_{t=0}^{T-1} \gamma^t (\frac{\nabla_{\theta_l} \pi(a_l^t, a_e^t | s^t)}{\pi(a_l^t, a_e^t | s^t)} J_{ed}(s^t, a^t)^T + \nabla_{\theta_l} r_{\theta_l}(s^t, a^t))],$$

$$= E^{\pi_{\theta_l, \theta_e}} [\sum_{t=0}^{T-1} \gamma^t (\nabla_{\theta_l} \ln(\pi_{\theta_l, \theta_e}) J_{ed}(s^t, a^t)^T + \nabla_{\theta_l} r_{\theta_l}(s^t, a^t))],$$

$$\overset{(vii)}{=} E^{\pi_{\theta_l, \theta_e}} [\sum_{t=0}^{T-1} \gamma^t [(\mu_l(s^t, a^t) - \mu_l(s^t)) J_{ed}(s^t, a^t)^T]] + \mu_l(\pi_{\theta_l, \theta_e}),$$

where $(vii)$ use the expression of the equation (12).
Similarly, we can get $\nabla_{\theta_l} J_{ei}(\pi_{\theta_l, \theta_e})$ as follows:

$$\nabla_{\theta_l} J_{ei}(\pi_{\theta_l, \theta_e}),$$

$$\overset{(i)}{=} \int_{s^0 \in S} P_0(s^0) \int_{a_l^0 \in a_l} \int_{a_e^0 \in a_e} [\nabla_{\theta_l} \pi(a_l^0, a_e^0 | s^0) J_{ei}(s^0, a_l^0, a_e^0)$$

$$+ \pi(a_l^0, a_e^0 | s^0)(\nabla_{\theta_l} J_{ei}(s^0, a_l^0, a_e^0))] da_l^0 da_e^0 ds^0,$$

$$\overset{(viii)}{=} \int_{s^0 \in S} P_0(s^0) \int_{a_l^0 \in a_l} \int_{a_e^0 \in a_e} [\pi(a_l^0, a_e^0 | s^0) \nabla_{\theta_l} \ln(\pi(a_l^0, a_e^0 | s^0)) J_{ei}(s^0, a_l^0, a_e^0)$$

$$+ \pi(a_l^0, a_e^0 | s^0) \gamma \int_{s^1 \in S} P(s^1 | s^0, a_l^0, a_e^0) \nabla_{\theta_l} J_{ei}(s^1) ds^1] da_l^0 da_e^0 ds^0,$$

$$\overset{(vii)}{=} E^{\pi_{\theta_l, \theta_e}} [\sum_{t=0}^{T-1} \gamma^t (\mu_l(s^t, a^t) - \mu_l(s^t)) J_{ei}(s^t, a^t)^T],$$

where $(vii)$ use the expression of the equation (12).
By summing the result of $\nabla_{\theta_l} J_{ed}(\pi_{\theta_l, \theta_e})$ and $\nabla_{\theta_l} J_{ei}(\pi_{\theta_l, \theta_e})$, the result of $\nabla_{\theta_l} f(\theta_l, \theta_e)$ is as follows:

$$\nabla_{\theta_l} f(\theta_l, \theta_e),$$

$$= E^{\pi_{\theta_l, \theta_e}} [\sum_{t=0}^{T-1} \gamma^t [(\mu_l(s^t, a^t) - \mu_l(s^t)) J_{ed}(s^t, a^t)^T]] + \mu_l(\pi_{\theta_l, \theta_e})$$

$$+ E^{\pi_{\theta_l, \theta_e}} [\sum_{t=0}^{T-1} \gamma^t (\mu_l(s^t, a^t) - \mu_l(s^t)) J_{ei}(s^t, a^t)^T],$$

$$\overset{(ii)}{=} E^{\pi_{\theta_l, \theta_e}} [\sum_{t=0}^{T-1} \gamma^t [(\mu_l(s^t, a^t) - \mu_l(s^t))(J_{ed}(s^t, a^t) + J_{ei}(s^t, a^t))^T]] + \mu_l(\pi_{\theta_l, \theta_e}).$$

The $\nabla_{\theta_e} f(\theta_l, \theta_e)$ is calculated in the same way as the $\nabla_{\theta_l} f(\theta_l, \theta_e)$.

The process for getting $\nabla^2_{\theta_e} L(\theta_l, \theta_e)$ is shown below.

As we proved in Lemma 1, the gradient $\nabla_{\theta_e} L(\theta_l, \theta_e) = \mu_e(\pi_{\theta_l, \theta_e}) - \hat{\mu}_e(D_{\theta_l, r_e}) + \lambda\theta_e$, as a result, we can take the derivative of each term separately.

The derivative of $\mu_e(\pi_{\theta_l, \theta_e})$ w.r.t $\theta_e$ is calculated as follows:

$$\nabla_{\theta_e} \mu_e(\pi_{\theta_l, \theta_e}),$$

$$\overset{(i)}{=} \int_{s^0 \in S} P_0(s^0) \int_{a_l^0 \in a_l} \int_{a_e^0 \in a_e} [\nabla_{\theta_e} \pi(a_l^0, a_e^0|s^0)\mu_e(s^0, a_l^0, a_e^0)^T$$

$$+ \pi(a_l^0, a_e^0|s^0)\nabla_{\theta_e}\mu_e(s^0, a_l^0, a_e^0)^T]da_l^0 da_e^0 ds^0,$$

$$= \int_{s^0 \in S} P_0(s^0) \int_{a_l^0 \in a_l} \int_{a_e^0 \in a_e} [\nabla_{\theta_l} \pi(a_l^0, a_e^0|s^0)\mu_e(s^0, a_l^0, a_e^0)^T$$

$$+ \pi(a_l^0, a_e^0|s^0)(\nabla^2_{\theta_e} r_e^{\theta_e}(s^0, a_l^0, a_e^0) + \gamma \int_{s^1 \in S} P(s^1|s^0, a_l^0, a_e^0)\nabla_{\theta_e}\mu_e(s^1))ds^1]da_l^0 da_e^0 ds^0,$$

$$\overset{(viii)}{=} \int_{s^0 \in S} P_0(s^0) \int_{a_l^0 \in a_l} \int_{a_e^0 \in a_e} \{\nabla_{\theta_l} \pi(a_l^0, a_e^0|s^0)\mu_e(s^0, a_l^0, a_e^0)^T$$

$$+ \pi(a_l^0, a_e^0|s^0)[\nabla^2_{\theta_e} r_e^{\theta_e}(s^0, a_l^0, a_e^0)$$

$$+ \gamma \int_{s^1 \in S} P(s^1|s^0, a_l^0, a_e^0) \int_{a_l^1 \in a_l} \int_{a_e^1 \in a_e} \nabla_{\theta_e} \pi(a_l^1, a_e^1|s^1)\mu_e(s^1, a_l^1, a_e^1)^T$$

$$+ \pi(a_l^1, a_e^1|s^1)(\nabla^2_{\theta_e} r_e^{\theta_e}(s^1, a_l^1, a_e^1) + \gamma \int_{s^2 \in S} P(s^2|s^1, a_l^1, a_e^1)\nabla_{\theta_e}\mu_e(s^2))ds^2]ds^1\}da_l^0 da_e^0 ds^0,$$

$$= E^{\pi_{\theta_l, \theta_e}} \left[\sum_{t=0}^{T-1} \gamma^t \left(\frac{\nabla_{\theta_e} \pi(a_l^t, a_e^t|s^t)}{\pi(a_l^t, a_e^t|s^t)}\mu_e(s^t, a^t)^T + \nabla^2_{\theta_e} r_e^{\theta_e}(s^t, a^t)\right)\right],$$

$$= E^{\pi_{\theta_l, \theta_e}} \left[\sum_{t=0}^{T-1} \gamma^t (\nabla_{\theta_e} \ln(\pi_{\theta_l, \theta_e})\mu_e(s^t, a^t)^T + \nabla^2_{\theta_e} r_e^{\theta_e}(s^t, a^t))\right],$$

$$\overset{(vii)}{=} E^{\pi_{\theta_l, \theta_e}} \left[\sum_{t=0}^{T-1} \gamma^t [(\mu_e(s^t, a^t) - \mu_e(s^t))\mu_e(s^t, a^t)^T + \nabla^2_{\theta_e} r_e^{\theta_e}(s^t, a^t)]\right],$$

where $(vii)$ use the expression of the equation (12).

With the result of $\nabla_{\theta_e} \mu_e(\pi_{\theta_l, \theta_e})$, the derivative of $\nabla_{\theta_e} L(\theta_l, \theta_e)$ w.r.t $\theta_e$ is as follows:

$$\nabla^2_{\theta_e} L(\theta_l, \theta_e) = \nabla_{\theta_e}(\nabla_{\theta_e} L(\theta_l, \theta_e)),$$

$$= \nabla_{\theta_e}(\mu_e(\pi_{\theta_l, \theta_e}) - \hat{\mu}_e(D_{\theta_l, r_e}) + \lambda\theta_e),$$

$$\overset{(ii)}{=} E^{\pi_{\theta_l, \theta_e}} \left[\sum_{t=0}^{T-1} \gamma^t [(\mu_e(s^t, a^t) - \mu_e(s^t))\mu_e(s^t, a^t)^T + \nabla^2_{\theta_e} r_e^{\theta_e}(s^t, a^t)]\right]$$

$$- \frac{1}{d} \sum_{i=0}^{d} \sum_{t=0}^{T-1} \gamma^t \nabla^2_{\theta_e} r_e^{\theta_e}(s^{it}, a^{it}) + \lambda.$$

Through the same process of calculating $\nabla_{\theta_e} \mu_e(\pi_{\theta_l, \theta_e})$, the result is as follows:

$$\nabla^2_{\theta_l \theta_e} L(\theta_l, \theta_e) = E^{\pi_{\theta_l, \theta_e}} \left[\sum_{t=0}^{T-1} \gamma^t (\mu_e(s, a) - \mu_e(s))\mu_l(s, a)^T\right]$$

$\square$

## A.5 PROPERTIES OF THE LOWER LEVEL OPTIMIZATION PROBLEM

**Lemma 4.** *Suppose Assumption 1 holds, for any $\theta_l \in \mathbb{R}^n$ and $\theta_e \in \mathbb{R}^m$, $L$ is continuously twice differentiable in $(\theta_l, \theta_e)$.*

*For any $\bar{\theta}_1 \in \mathbb{R}^n$, $\nabla_{\theta_e} L(\bar{\theta}_1, \theta_e)$ is Lipschitz continuous (w.r.t $\theta_e$) with constant $L_{L_{\theta_e}} > 0$.*

*For any $\bar{\theta}_1 \in \mathbb{R}^n$ and $\bar{\theta}_2 \in \mathbb{R}^m$, we have $\|\nabla^2_{\theta_l \theta_e} L(\bar{\theta}_1, \bar{\theta}_2)\| \leq C_{L_{\theta_l \theta_e}}$ for some constant $C_{L_{\theta_l \theta_e}} > 0$.*

*For any $\bar{\theta}_1 \in \mathbb{R}^n$, $\nabla^2_{\theta_l \theta_e} L(\bar{\theta}_1, \theta_e)$ and $\nabla^2_{\theta_e \theta_e} L(\bar{\theta}_1, \theta_e)$ are Lipschitz continuous (w.r.t $\theta_e$) with constants $L_{L_{\theta_l \theta_e}} > 0$ and $L_{L_{\theta_e \theta_e}} > 0$.*

*For any $\bar{\theta}_2 \in \mathbb{R}^m$, $\nabla^2_{\theta_l \theta_e} L(\theta_l, \bar{\theta}_2)$ and $\nabla^2_{\theta_e \theta_e} L(\theta_l, \bar{\theta}_2)$ are Lipschitz continuous (w.r.t $\theta_l$) with constants $\bar{L}_{L_{\theta_l \theta_e}} > 0$ and $\bar{L}_{L_{\theta_e \theta_e}} > 0$*

*Proof.* Suppose that $h$ is a real-valued function defined and differentiable on an interval $H \subset r_n$. If $\|\nabla h\|$ is bounded on I, then $h$ is a Lipschitz function on $H$. So we need to prove $\nabla^2_{\theta_e \theta_e} L(\theta_l, \theta_e)$ is bounded. From Assumption 1, we can show that $\exists R_{1g} > 0, \|\nabla r_{\theta_l}\| \leq R_{1g}$.

$$\|\mu_l(s)\| \leq E^{\pi_{\theta_l, \theta_e}} \Big[ \sum_{t=0}^{T-1} \gamma^t R_{1g} | s^0 = s \Big] \overset{(i)}{\leq} \frac{R_{1g}}{1 - \gamma},$$

where $(i)$ uses the close form of geometric series.

As a result, $\|\mu_l(s)\|$ is bounded, proved through the same way, $\|\mu_e(s)\|, \|\mu_l(s, a)\|, \|\mu_e(s, a)\|$ are also bounded. Based on the Lemma 3, all elements of $\nabla^2_{\theta_e \theta_e} L(\theta_l, \theta_e)$ are finite, therefore, $\|\nabla^2_{\theta_e \theta_e} L(\theta_l, \theta_e)\|$ is bounded and the $\nabla_{\theta_e} L(\bar{\theta}_1, \theta_e)$ is Lipschitz continuous.

In the same way of proving $\|\nabla^2_{\theta_e \theta_e} L(\theta_l, \theta_e)\|$ is bounded, we can show $\|\nabla^2_{\theta_l \theta_e} L(\theta_l, \theta_e)\|$ is bounded.

We need to prove $\nabla^3_{\theta_e \theta_e \theta_e} L(\theta_l, \theta_e)$ and $\nabla^3_{\theta_l \theta_e \theta_e} L(\theta_l, \theta_e)$ are bounded. The proof of $\nabla^3_{\theta_l \theta_e \theta_e} L(\theta_l, \theta_e)$ is bounded as follows:

$$\nabla^3_{\theta_l \theta_e \theta_e} L(\theta_l, \theta_e),$$

$$= \nabla_{\theta_e} \Big( E^{\pi_{\theta_l, \theta_e}} \Big[ \sum_{t=0}^{T-1} \gamma^t (\mu_e(s, a) - \mu_e(s)) \mu_l(s, a)^T \Big] \Big),$$

$$\overset{(i)}{=} E^{\pi_{\theta_l, \theta_e}} \Big[ \sum_{t=0}^{T-1} \gamma^t (\nabla_{\theta_e} \mu_e(s, a)) \mu_l(s, a)^T) + \mu_e(s, a)(\nabla_{\theta_e} \mu_l(s, a)^T)$$

$$- (\nabla_{\theta_e} \mu_e(s)) \mu_l(s, a)^T - \mu_e(s)(\nabla_{\theta_e} \mu_l(s, a)^T) \Big].$$

Each gradient inside the expectation could be derived through the process of deriving $\nabla_{\theta_l} \mu_l(\pi_{\theta_l, \theta_e})$ in the proof of Lemma 3 and these gradients are all finite with the same way of proving $\|\nabla^2_{\theta_e \theta_e} L(\theta_l, \theta_e)\|$ is bounded. The third-order gradient $\nabla^3_{\theta_l \theta_e \theta_e} L(\theta_l, \theta_e)$ is bounded with the smae process. Therefore, the third-order gradients of $L(\theta_l, \theta_e)$ are all bounded.

Through the same procedure, we can prove $\nabla^3_{\theta_e \theta_e \theta_l} L(\theta_l, \theta_e)$ and $\nabla^3_{\theta_l \theta_e \theta_l} L(\theta_l, \theta_e)$ are bounded.

At the same time, the existence of $\nabla^2_{\theta_l \theta_e} L(\theta_l, \theta_e), \nabla^2_{\theta_e \theta_e} L(\theta_l, \theta_e)$ are shown. Analogously, the existence of $\nabla^2_{\theta_l \theta_l} L(\theta_l, \theta_e), \nabla^2_{\theta_e \theta_l} L(\theta_l, \theta_e)$ could be proved in the same way. The third-order gradients of $L(\theta_l, \theta_e)$ are bounded. Therefore, $L$ is continuously twice differentiable in $(\theta_l, \theta_e)$

$\square$

## A.6 PROPERTIES OF THE UPPER-LEVEL OPTIMIZATION PROBLEM

**Lemma 5.** *Suppose Assumption 1 holds, for any $\bar{\theta}_1 \in \mathbb{R}^n$, $\nabla_{\theta_l} f(\bar{\theta}_1; \theta_e)$ and $\nabla_{\theta_e} f(\bar{\theta}_1; \theta_e)$ are Lipschitz continuous (w.r.t $\theta_e$) with constants $L_{f_{\theta_l}} > 0$ and $L_{f_{\theta_e}} > 0$.*

*For any $\bar{\theta}_2 \in \mathbb{R}^m$, $\nabla_{\theta_e} f(\theta_l; \bar{\theta}_2)$ is Lipschitz continuous (w.r.t $\theta_l$) with constants $\bar{L}_{f_{\theta_e}} > 0$*

*For any $\bar{\theta}_1 \in \mathbb{R}^n$ and $\bar{\theta}_2 \in \mathbb{R}^m$, we have $\|\nabla_{\theta_e} f(\bar{\theta}_1; \bar{\theta}_2)\| \leq C_{f_{\theta_e}}$ for some $C_{f_{\theta_e}} > 0$.*

*Proof.*

$$\|J_{ed}(\pi_{\theta_l,\theta_e})\| \leq E^{\pi_{\theta_l,\theta_e}}\big[\sum_{t=0}^{T-1}\gamma^t r_{ld}|s^0 = s\big] \overset{(i)}{\leq} \frac{r_{ld}}{1-\gamma},$$

where $(i)$ uses the close form of geometric series.

So the cumulative expert-dependent reward value $J_{ed}$ is bounded, analogously, the cumulative expert-independent reward value $J_{ei}$ is bounded.

$$\nabla_{\theta_l\theta_e}f(\theta_l;\theta_e),$$

$$= \nabla_{\theta_e}(E^{\pi_{\theta_l,\theta_e}}\big[\sum_{t=0}^{T-1}\gamma^t[(\mu_l(s^t,a^t) - \mu_l(s^t))(J_{ed}(s^t,a^t) + J_{ei}(s^t,a^t))^T]\big] + \mu_l(\pi_{\theta_l,\theta_e}),$$

$$\overset{(i)}{=} E^{\pi_{\theta_l,\theta_e}}\big[\sum_{t=0}^{T-1}\gamma^t[(\nabla_{\theta_e}\mu_l(s^t,a^t) - \nabla_{\theta_e}\mu_l(s^t))(J_{ed}(s^t,a^t) + J_{ei}(s^t,a^t))^T$$

$$+ (\mu_l(s^t,a^t) - \mu_l(s^t))(\nabla_{\theta_e}J_{ed}(s^t,a^t) + \nabla_{\theta_e}J_{ei}(s^t,a^t))^T]\big] + \nabla_{\theta_e}\mu_l(\pi_{\theta_l,\theta_e}).$$

Refer to the proof of Lemma 4, all elements in $\nabla_{\theta_l\theta_e}f(\theta_l;\theta_e)$ are finite, itself is bounded. Analogously, $\nabla_{\theta_e\theta_e}f(\theta_l;\theta_e)$ is bounded under the same proofing process.

Through the same procedure , we can prove $\nabla_{\theta_e\theta_l}f(\theta_l;\theta_e)$ is bounded.

All element of $\nabla_{\theta_e}f(\bar{\theta}_1;\bar{\theta}_2)$ are bounded. Therefore, all elements for $\nabla_{\theta_e}f(\theta_l;\theta_e)$ is bounded and it is bounded.

$\square$

## A.7 PROPERTIES OF THE APPROXIMATION ERRORS

Define $b_{12}(k) = \hat{\nabla}^2_{\theta_l\theta_e}L(\theta_l(k),\theta_e(k)) - \nabla^2_{\theta_l\theta_e}L(\theta_l(k),\theta_e(k))$, $b_{22}(k) = \hat{\nabla}^2_{\theta_e\theta_e}L(\theta_l(k),\theta_e(k)) - \nabla^2_{\theta_e\theta_e}L(\theta_l(k),\theta_e(k))$, $b_1(k) = \hat{\nabla}_{\theta_l}f(\theta_l(k),\theta_e(k)) - \nabla_{\theta_l}f(\theta_l(k),\theta_e(k))$, $b_2(k) = \hat{\nabla}_{\theta_e}f(\theta_l(k),\theta_e(k)) - \nabla_{\theta_e}f(\theta_l(k),\theta_e(k))$, $b(k) = [\hat{\nabla}^2_{\theta_e\theta_e}L(\theta_l(k),\theta_e(k))]^{-1}\hat{\nabla}_{\theta_e}f(\theta_l(k),\theta_e(k)) - [\nabla^2_{\theta_e\theta_e}L(\theta_l(k),\theta_e(k))]^{-1}\nabla_{\theta_e}f(\theta_l(k),\theta_e(k))$ at iteration $k$, $b_a(k) = \hat{\nabla}f(\theta_l(k),\theta_e(k)) - \nabla f(\theta_l(k),\theta_e(k))$ at the iteration $k$. Through the same procedure in the proof of the Lemma 5, we can get conclude that each element of forth-order gradient of $L(\theta_l,\theta_e)$ is bounded by the constant $C_L$ and each element of the third-order gradient of $f(\theta_l(k),\theta_e(k))$ is bounded by $C_f$.

**Lemma 6.** *The biases $b_{12}(k)$, $b_{22}(k)$, $b_1(k)$, $b_2(k)$, $b(k)$, and $b_a(k)$ are bounded,*

$$\|b_{12}(k)\| \leq \frac{C_L p^2(k)\sqrt{m}}{6}\{[m^3 - (m-1)^3]\alpha_l^2 + (m-1)^3\alpha_l\alpha_0^3\},$$

$$\|b_{22}(k)\| \leq \frac{C_L p^2(k)\sqrt{m}}{6}\{[m^3 - (m-1)^3]\alpha_l^2 + (m-1)^3\alpha_l\alpha_0^3\},$$

$$\|b_1(k)\| \leq \frac{C_f p^2(k)\sqrt{n}}{6}\{[n^3 - (n-1)^3]\alpha_l^2 + (n-1)^3\alpha_l\alpha_0^3\},$$

$$\|b_2(k)\| \leq \frac{C_f p^2(k)\sqrt{m}}{6}\{[m^3 - (m-1)^3]\alpha_l^2 + (m-1)^3\alpha_l\alpha_0^3\},$$

$$\|b(k)\| \leq \frac{2C_{f_{\theta_e}}(C_L p^2(k) + C_f p^2(k))}{6\sqrt{m}\lambda^2}\{[m^3 - (m-1)^3]\alpha_l^2 + (m-1)^3\alpha_l\alpha_0^3\},$$

$$\|b_a(k)\|$$

$$\leq \frac{C_f p^2(k)\sqrt{n}}{6}\{[n^3 - (n-1)^3]\alpha_l^2 + (n-1)^3\alpha_l\alpha_0^3\}$$

$$+ \frac{2C_{L_{\theta_l\theta_e}}C_L p^2(k) + 2C_{L_{\theta_l\theta_e}}\sqrt{m}\lambda C_f p^2(k)}{6\sqrt{m}\lambda^2}$$

$$\{[m^3 - (m-1)^3]\alpha_l^2 + (m-1)^3\alpha_l\alpha_0^3\} + \frac{C_{f_{\theta_e}}C_L p^2(k)}{6m\lambda^3}\{[m^3 - (m-1)^3]\alpha_l^2 + (m-1)^3\alpha_l\alpha_0^3\}$$

$$+ \frac{2p^4(k)C_L(C_{f_{\theta_e}}C_L + \sqrt{m}\lambda C_f)}{36\lambda^2}\{[m^3 - (m-1)^3]\alpha_l^2 + (m-1)^3\alpha_l\alpha_0^3\}^2,$$

*Proof.* According to the Lemma 1 in (Spall, 1992), the approximation error $b_{12l}(k)$ for $\hat{\nabla}^2_{\theta_l \theta_e} L(\theta_l, \theta_e)$ is :

$$b_{12l}(k) \leq \frac{C_L p^2(k)}{6}\{[m^3 - (m-1)^3]\alpha_l^2 + (m-1)^3 \alpha_l \alpha_0^3\},$$

where $b_{12l}(k)$ represent the $l$th term of the bias $b_{12l}(k)$ at $k$th iteration.

$$\|b_{12}(k)\| \leq \frac{C_L p^2(k)\sqrt{m}}{6}\{[m^3 - (m-1)^3]\alpha_l^2 + (m-1)^3 \alpha_l \alpha_0^3\}.$$

Analogously, the $b_{22}(k)$, $b_1(k)$ and $b_2(k)$ are also bounded.

With the $\hat{\nabla}^2_{\theta_e \theta_e} L(\theta_l(k), \theta_e(k))$ and $\hat{\nabla}_{\theta_e} f(\theta_l(k), \theta_e(k))$, we estimate the $[\nabla^2_{\theta_e \theta_e} L(\theta_l(k), \theta_e(k))]^{-1} \nabla_{\theta_e} f(\theta_l(k), \theta_e(k))$ through conjugate gradient.

$$\|\hat{\nabla}^2_{\theta_e \theta_e} L(\theta_l(k), \theta_e(k))\| = \|\nabla^2_{\theta_e \theta_e} L(\theta_l(k), \theta_e(k)) + b_{22}(k)\| \geq \|\lambda I + b_{22}(k)\|.$$

Then we can tune the parameters of $b_{22}(k)$ such as $\Delta, \alpha_0$ and $\alpha_l$ to let $\|[\hat{\nabla}^2_{\theta_e \theta_e} L(\theta_l, \theta_e)]^{-1}\| \leq \frac{2}{\sqrt{m}\lambda}$ and make sure the $\hat{\nabla}^2_{\theta_e \theta_e} L(\theta_l, \theta_e) \geq \frac{\lambda}{2} I$

$$\|b(k)\|,$$
$$= \|E[[\hat{\nabla}^2_{\theta_e \theta_e} L(\theta_l(k), \theta_e(k))]^{-1} \hat{\nabla}_{\theta_e} f(\theta_l(k), \theta_e(k))$$
$$- [\nabla^2_{\theta_e \theta_e} L(\theta_l(k), \theta_e(k))]^{-1} \nabla_{\theta_e} f(\theta_l(k), \theta_e(k))]\|,$$
$$\overset{(iv)}{\leq} \|E[\hat{\nabla}^2_{\theta_e \theta_e} L(\theta_l(k), \theta_e(k))]^{-1} (\hat{\nabla}^2_{\theta_e \theta_e} L(\theta_l(k), \theta_e(k)) - \nabla^2_{\theta_e \theta_e} L(\theta_l(k), \theta_e(k)))$$
$$[\nabla^2_{\theta_e \theta_e} L(\theta_l(k), \theta_e(k))]^{-1}]\|\|\nabla_{\theta_e} f(\theta_l(k), \theta_e(k))\| + \|\hat{\nabla}^2_{\theta_e \theta_e} L(\theta_l(k), \theta_e(k))]^{-1} b_2(k)\|,$$
$$\overset{(v)}{\leq} \|[\hat{\nabla}^2_{\theta_e \theta_e} L(\theta_l(k), \theta_e(k))]^{-1}\|\|\hat{\nabla}^2_{\theta_e \theta_e} L(\theta_l(k), \theta_e(k)) - \nabla^2_{\theta_e \theta_e} L(\theta_l(k), \theta_e(k))\|\|$$
$$[\nabla^2_{\theta_e \theta_e} L(\theta_l(k), \theta_e(k))]^{-1}\|\|\nabla_{\theta_e} f(\theta_l(k), \theta_e(k))\| + \|\hat{\nabla}^2_{\theta_e \theta_e} L(\theta_l(k), \theta_e(k))]^{-1}\|\|b_2 k\|,$$
$$\overset{(vii)}{\leq} \frac{2C_{f_{\theta_e}}\|b_{22}(k)\| + 2\sqrt{m}\lambda\|b_2(k)\|}{m\lambda^2},$$
$$\leq \frac{2C_{f_{\theta_e}}(C_L p^2(k) + C_f p^2(k))}{6\sqrt{m}\lambda^2}\{[m^3 - (m-1)^3]\alpha_l^2 + (m-1)^3 \alpha_l \alpha_0^3\},$$

where $(vii)$ uses the result of Lemma (5).

$$\|b_a(k)\| = E[\|\hat{\nabla} f(\theta_l(k), \theta_e(k))\|] - \|\nabla f(\theta_l(k), \theta_e(k))\|,$$
$$\overset{(vii)}{=} E[\|\hat{\nabla}_{\theta_l} f(\theta_l(k), \theta_e(k)) - \hat{\nabla}^2_{\theta_l \theta_e} L(\theta_l(k), \theta_e(k))[\hat{\nabla}^2_{\theta_e \theta_e} L(\theta_l(k), \theta_e(k))]^{-1}$$
$$\hat{\nabla}_{\theta_e} f(\theta_l(k), \theta_e(k))\|] - \|\nabla f(\theta_l(k), \theta_e(k))\|,$$
$$= \|\nabla_{\theta_l} f(\theta_l(k), \theta_e(k)) + b_1(k) - (\nabla^2_{\theta_l \theta_e} L(\theta_l(k), \theta_e(k)) + b_{12}(k))$$
$$([\nabla^2_{\theta_e \theta_e} L(\theta_l(k), \theta_e(k))]^{-1} \nabla_{\theta_e} f(\theta_l(k), \theta_e(k)) + b(k))\| - \|\nabla f(\theta_l(k), \theta_e(k))\|$$
$$\overset{(v)}{\leq} \|b_1(k)\| + \|\nabla^2_{\theta_l \theta_e} L(\theta_l(k), \theta_e(k))\|\|b(k)\|$$
$$+ \|[\nabla^2_{\theta_e \theta_e} L(\theta_l(k), \theta_e(k))]^{-1}\|\|\nabla_{\theta_e} f(\theta_l(k), \theta_e(k))\|\|b_{12}(k)\| + \|b_{12}(k)\|\|b(k)\|,$$
$$\overset{(vii)}{\leq} \frac{C_f p^2(k)\sqrt{n}}{6}\{[n^3 - (n-1)^3]\alpha_l^2 + (n-1)^3 \alpha_l \alpha_0^3\}$$
$$+ \frac{2C_{L_{\theta_l \theta_e}} C_L p^2(k) + 2C_{L_{\theta_l \theta_e}} \sqrt{m}\lambda C_f p^2(k)}{6\sqrt{m}\lambda^2}\{[m^3 - (m-1)^3]\alpha_l^2 + (m-1)^3 \alpha_l \alpha_0^3\}$$
$$+ \frac{C_{f_{\theta_e}} C_L p^2(k)}{6m\lambda^3}\{[m^3 - (m-1)^3]\alpha_l^2 + (m-1)^3 \alpha_l \alpha_0^3\}$$
$$+ \frac{2p^4(k)C_L(C_{f_{\theta_e}} C_L + \sqrt{m}\lambda C_f)}{36\lambda^2}\{[m^3 - (m-1)^3]\alpha_l^2 + (m-1)^3 \alpha_l \alpha_0^3\}^2,$$

where the first $(vii)$ uses the result of Lemma (2), and the second $(vii)$ uses the result of Lemma (5). $\qquad\square$

### A.7.1 PROOF OF THEOREM 2

From the Lemma 2.2 of (Ghadimi & Wang, 2018), $\|\nabla f(\theta_l(k), \theta_e(k)) - \nabla f(\theta_l(k), \theta_e^*(\theta_l(k)))\| \leq C\|\theta_e^*(\theta_l(k)) - \theta_e(k)\|$, $\|\nabla f(\theta_l(k'), \theta_e^*(\theta_l(k'))) - \nabla f(\theta_l(k), \theta_e^*(\theta_l(k)))\| \leq L_f \|\theta_l(k') - \theta_l(k)\|$

where $C = L_{f_{\theta_l}} + \frac{L_{f_{\theta_e}} C_{L_{\theta_l \theta_e}}}{\lambda} + C_{f_{\theta_e}} [\frac{L_{L_{\theta_l \theta_e}}}{\lambda} + \frac{L_{L_{\theta_e \theta_e}} C_{L_{\theta_l \theta_e}}}{\lambda^2}], L_f = \frac{(\bar{L}_{f_{\theta_e}} + C) C_{L_{\theta_l \theta_e}}}{\lambda} + L_{f_{\theta_l}} + C_{f_{\theta_e}} [\frac{\bar{L}_{L_{\theta_l \theta_e}} C_{f_{\theta_e}}}{\lambda} + \frac{\bar{L}_{L_{\theta_e \theta_e}} C_{L_{\theta_l \theta_e}}}{\lambda^2}], Q_L = \frac{L_{L_{\theta_e}}}{\lambda}$ denotes the condition number of $L(\theta_l, \theta_e)$, $M = \max_{\theta_l \in \theta_l} \|\theta_e(0) - \theta_e^*(\theta_l)\|$, $D_{\theta_l} = \max_{x,y \in \theta_l} \{\|x - y\|\}$.

The proof of Theorem 2 is as follows:

*Proof.* First we compute the variance of $\hat{\nabla} f(\theta_l(k), \theta_e(k))$.

$\|\hat{\nabla} f(\theta_l(k), \theta_e(k))\|$,

$\overset{(vii)}{=} \|\hat{\nabla}_{\theta_l} f(\theta_l(k), \theta_e(k)) - \hat{\nabla}^2_{\theta_l \theta_e} L(\theta_l(k), \theta_e(k)) [\hat{\nabla}^2_{\theta_e \theta_e} L(\theta_l(k), \theta_e(k))]^{-1} \hat{\nabla}_{\theta_e} f(\theta_l(k), \theta_e(k))\|$,

$= \|\nabla_{\theta_l} f(\theta_l(k), \theta_e(k)) + b_1(k) - (\nabla^2_{\theta_l \theta_e} L(\theta_l(k), \theta_e(k)) + b_{12}(k))$

$([\nabla^2_{\theta_e \theta_e} L(\theta_l(k), \theta_e(k))]^{-1} \nabla_{\theta_e} f(\theta_l(k), \theta_e(k)) + b(k))\|$,

$\overset{(v)}{\leq} \|\nabla_{\theta_l} f(\theta_l(k), \theta_e(k))\| + \|b_1(k)\| + \|\nabla^2_{\theta_l \theta_e} L(\theta_l(k), \theta_e(k))\| \|[\nabla^2_{\theta_e \theta_e} L(\theta_l(k), \theta_e(k))]^{-1}\|$

$\|\nabla_{\theta_e} f(\theta_l(k), \theta_e(k))\| + \|\nabla^2_{\theta_l \theta_e} L(\theta_l(k), \theta_e(k))\| \|b(k)\| + \|b_1(k)\| \|[\nabla^2_{\theta_e \theta_e} L(\theta_l(k), \theta_e(k))]^{-1}\|$

$\|\nabla_{\theta_e} f(\theta_l(k), \theta_e(k))\| + \|b_1(k)\| \|b(k)\|$,

$\overset{(vii)}{\leq} C_{f_{\theta_l}} + \frac{C_f p^2(k) \sqrt{n}}{6} \{[n^3 - (n-1)^3] \alpha_l^2 + (n-1)^3 \alpha_l \alpha_0^3\}$

$+ \frac{C_{L_{\theta_l \theta_e}} C_{f_{\theta_e}}}{\sqrt{m} \lambda} + \frac{2 C_{L_{\theta_l \theta_e}} C_{f_{\theta_e}} (C_L p^2(k) + C_f p^2(k))}{6\sqrt{m} \lambda^2} \{[m^3 - (m-1)^3] \alpha_l^2 + (m-1)^3 \alpha_l \alpha_0^3\}$

$+ \frac{C_{f_{\theta_e}}}{\sqrt{m} \lambda} \frac{C_f p^2(k) \sqrt{n}}{6} \{[n^3 - (n-1)^3] \alpha_l^2 + (n-1)^3 \alpha_l \alpha_0^3\}$

$+ \frac{2 C_f p^2(k) \sqrt{n} C_{f_{\theta_e}} (C_L p^2(k) + C_f p^2(k))}{36\sqrt{m} \lambda^2}$

$\{[n^3 - (n-1)^3] \alpha_l^2 + (n-1)^3 \alpha_l \alpha_0^3\} \{[m^3 - (m-1)^3] \alpha_l^2 + (m-1)^3 \alpha_l \alpha_0^3\}$,

where the first $(vii)$ uses the result of Lemma (2), and the second $(vii)$ uses the result of Lemma (5) and Lemma (6).

Since $\hat{\nabla} f(\theta_l(k), \theta_e(k))$ is bounded,

$Var(\hat{\nabla} f(\theta_l(k), \theta_e(k)))$,

$\leq (C_{f_{\theta_l}} + \frac{C_f p^2(k) \sqrt{n}}{6} \{[n^3 - (n-1)^3] \alpha_l^2 + (n-1)^3 \alpha_l \alpha_0^3\}$

$+ \frac{C_{L_{\theta_l \theta_e}} C_{f_{\theta_e}}}{\sqrt{m} \lambda} + \frac{2 C_{L_{\theta_l \theta_e}} C_{f_{\theta_e}} (C_L p^2(k) + C_f p^2(k))}{6\sqrt{m} \lambda^2} \{[m^3 - (m-1)^3] \alpha_l^2 + (m-1)^3 \alpha_l \alpha_0^3\}$

$+ \frac{C_{f_{\theta_e}}}{\sqrt{m} \lambda} \frac{C_f p^2(k) \sqrt{n}}{6} \{[n^3 - (n-1)^3] \alpha_l^2 + (n-1)^3 \alpha_l \alpha_0^3\}$

$+ \frac{2 C_f p^2(k) \sqrt{n} C_{f_{\theta_e}} (C_L p^2(k) + C_f p^2(k))}{36\sqrt{m} \lambda^2}$

$\{[n^3 - (n-1)^3] \alpha_l^2 + (n-1)^3 \alpha_l \alpha_0^3\} \{[m^3 - (m-1)^3] \alpha_l^2 + (m-1)^3 \alpha_l \alpha_0^3\})^2$.

Then we need to find the total bias, the total bias $b_t(k)$ is the sum of the bias from approximation and $\nabla f(\theta_l(k), \theta_e(k)) - \nabla f(\theta_l(k), \theta_e^*(\theta_l(k)))$

$$\|b_t(k)\|,$$
$$= \|E[\nabla f(\theta_l(k), \theta_e(k))] - \nabla f(\theta_l(k), \theta_e^*(k))\|,$$
$$\overset{(vii)}{\leq} \|b_a(k)\| + \|r_{li}(\frac{Q_L-1}{Q_L+1})^{t_k}\|\theta_e(0) - \theta_e^*(\theta_l(k))\|,$$

where $(vii)$ adds the approximation error $r_{li}(\frac{Q_L-1}{Q_L+1})^{t_k}\|\theta_e(0) - \theta_e^*(\theta_l(k))$ from the lower level. Next step is to find the bound for $E[\|\nabla f(\theta_l(k), \theta_e^*(\theta_l(k)))\|^2]$.

$$f(\theta_l(k+1), \theta_e^*(\theta_l(k+1))),$$
$$\overset{(vi)}{\leq} f(\theta_l(k), \theta_e^*(\theta_l(k))) + \langle \nabla f(\theta_l(k), \theta_e^*(\theta_l(k))), \theta_l(k+1) - \theta_l(k)\rangle + \frac{L_f}{2}\|\theta_l(k+1) - \theta_l(k)\|^2,$$
$$= f(\theta_l(k), \theta_e^*(\theta_l(k))) - \alpha_k\langle \nabla f(\theta_l(k), \theta_e^*(\theta_l(k))), \hat{\nabla} f(\theta_l(k), \theta_e(k))\rangle + \frac{L_f\alpha_k^2}{2}\|\hat{\nabla} f(\theta_l(k), \theta_e(k))\|^2.$$

The expectation of $f(\theta_l(k+1), \theta_e^*(\theta_l(k+1)))$ becomes:

$$E[f(\theta_l(k+1), \theta_e^*(\theta_l(k+1)))],$$
$$\leq f(\theta_l(k), \theta_e^*(\theta_l(k))) - \alpha_k\langle \nabla f(\theta_l(k), \theta_e^*(\theta_l(k))), \nabla f(\theta_l(k), \theta_e^*(\theta_l(k))) + b_t(k)\rangle$$
$$+ \frac{L_f\alpha_k^2}{2}E\|\nabla f(\theta_l(k), \theta_e^*(\theta_l(k))) + \hat{\nabla} f(\theta_l(k), \theta_e(k)) - \nabla f(\theta_l(k), \theta_e^*(\theta_l(k)))\|^2,$$
$$\leq f(\theta_l(k), \theta_e^*(\theta_l(k))) - \alpha_k\langle \nabla f(\theta_l(k), \theta_e^*(\theta_l(k))), \nabla f(\theta_l(k), \theta_e^*(\theta_l(k))) + b_t(k)\rangle$$
$$+ \frac{L_f\alpha_k^2}{2}Var(\hat{\nabla} f(\theta_l(k), \theta_e(k))) + \frac{L_f\alpha_k^2}{2}E\|\nabla f(\theta_l(k), \theta_e^*(\theta_l(k)))\|^2$$
$$+ L_f\alpha_k^2\langle \nabla f(\theta_l(k), \theta_e^*(\theta_l(k))), b_t(k)\rangle,$$
$$= f(\theta_l(k), \theta_e^*(\theta_l(k))) - (\alpha_k - \frac{L_f\alpha_k^2}{2})\|\nabla f(\theta_l(k), \theta_e^*(\theta_l(k)))\|^2$$
$$- (\alpha_k - L_f\alpha_k^2)\langle \nabla f(\theta_l(k), \theta_e^*(\theta_l(k))), b_t(k)\rangle + \frac{L_f\alpha_k^2}{2}Var(\hat{\nabla} f(\theta_l(k), \theta_e(k))) + \frac{L_f\alpha_k^2}{2}\|b_t(k)\|^2.$$

Choose $\alpha_k \leq \frac{1}{L_f}$ and with the fact $2\langle \nabla f(\theta_l(k), \theta_e^*(\theta_l(k))), b_t(k)\rangle \leq \|\nabla f(\theta_l(k), \theta_e^*(\theta_l(k)))\|^2 + \|b_t(k)\|^2$.

$$E[f(\theta_l(k+1), \theta_e^*(\theta_l(k+1)))],$$
$$\leq f(\theta_l(k), \theta_e^*(\theta_l(k))) - \frac{\alpha_k}{2}\|\nabla f(\theta_l(k), \theta_e^*(\theta_l(k)))\|^2$$
$$+ \frac{\alpha_k}{2}\|b_t(k)\|^2 + \frac{L_f\alpha_k^2}{2}Var(\hat{\nabla} f(\theta_l(k), \theta_e(k))),$$

Rearrange terms,

$$\sum_{k=0}^{K-1}\frac{\alpha_k}{2}E[\|\nabla f(\theta_l(k), \theta_e^*(\theta_l(k)))\|^2],$$
$$\leq f(\theta_l(0), \theta_e^*(\theta_l(0))) - f^* + \sum_{k=0}^{K-1}(\frac{\alpha_k}{2}\|b_t(k)\|^2 + \frac{L_f\alpha_k^2}{2}Var(\hat{\nabla} f(\theta_l(k), \theta_e(k)))).$$

For $\|b_t(k)\|^2$, it is the linear combination of $p^4(k), p^6(k), p^8(k), p^2(k)(\frac{Q_L-1}{Q_L+1})^{t_k}+, p^4(k)(\frac{Q_L-1}{Q_L+1})^{t_k}, (\frac{Q_L-1}{Q_L+1})^{2t_k}$. For $Var(\hat{\nabla} f(\theta_l(k), \theta_e(k))))$, it is the linear combination of $1, p^2(k), p^4(k), p^6(k), p^8(k)$. For simplification, we use $C_{si} > 0, i = 1, 2, \ldots$ to represent the constant for all combinations of terms involve $p(k)$, $\alpha_k$ and $t_k$. Then we can continue the calculation as follows:

$$\frac{1}{K} \sum_{k=0}^{K-1} E[\|\nabla f(\theta_l(k), \theta_e^*(\theta_l(k)))\|^2],$$

$$\leq \frac{2}{K\alpha_k} f(\theta_l(0), \theta_e^*(\theta_l(0))) - f^* + \frac{1}{K} \sum_{k=0}^{K-1} (\|b_t(k)\|^2 + L_f \alpha_k Var(\hat{\nabla} f(\theta_l(k), \theta_e(k)))),$$

$$\overset{(viii)}{\leq} \frac{2(f(\theta_l(0), \theta_e^*(\theta_l(0))) - f^*)}{\alpha_k K} + \frac{C_{s1}}{K} \sum_{k=0}^{K-1} p^4(k) + \frac{C_{s2}}{K} \sum_{k=0}^{K-1} p^6(k) + \frac{C_{s3}}{K} \sum_{k=0}^{K-1} p^8(k)$$

$$+ \frac{C_{s4}}{K} \sum_{k=0}^{K-1} p^2(k) (\frac{Q_L - 1}{Q_L + 1})^{t_k} + \frac{C_{s5}}{K} \sum_{k=0}^{K-1} p^4(k) (\frac{Q_L - 1}{Q_L + 1})^{t_k} + \frac{C_{s6}}{K} \sum_{k=0}^{K-1} (\frac{Q_L - 1}{Q_L + 1})^{2t_k}$$

$$+ \frac{C_{s7}}{K} \sum_{k=0}^{K-1} \alpha_k + \frac{C_{s8}}{K} \sum_{k=0}^{K-1} \alpha_k p^2(k) + \frac{C_{s9}}{K} \sum_{k=0}^{K-1} \alpha_k p^4(k) + \frac{C_{s10}}{K} \sum_{k=0}^{K-1} \alpha_k p^6(k)$$

$$+ \frac{C_{s11}}{K} \sum_{k=0}^{K-1} \alpha_k p^8(k),$$

where $(viii)$ expands each term of $\|b_t(k)\|^2$ and $Var(\hat{\nabla} f(\theta_l(k), \theta_e(k))))$. Choose $p(k) = \frac{1}{k}$, $\alpha_k = \frac{1}{L_f \sqrt{K}}$, $t_k = \lceil \frac{\sqrt[4]{k+1}}{2} \rceil$. Since $0 \leq \frac{Q_L - 1}{Q_L + 1} < 1$, we can conclude $\sum_{k=0}^{K-1} p^2(k) (\frac{Q_L - 1}{Q_L + 1})^{t_k} < \sum_{k=0}^{K-1} p^2(k)$ when $\frac{Q_L - 1}{Q_L + 1} \neq 0$, then the convergence rate is as follows:

$$\frac{1}{K} \sum_{k=0}^{K-1} E[\|\nabla f(\theta_l(k), \theta_e^*(\theta_l(k)))\|^2] \leq \frac{C_{s12}}{\sqrt{K}} + \frac{C_{s13}}{K} + \frac{C_{s14}}{K\sqrt{K}}.$$

As $K \to \infty$, $\frac{1}{K} \sum_{k=0}^{K-1} E[\|\nabla f(\theta_l(k), \theta_e^*(\theta_l(k)))\|^2] \to 0$, which shows that $E[\nabla f(\theta_l(k), \theta_e^*(\theta_l(k)))]$ decreases at the rate of $\mathcal{O}(\frac{1}{\sqrt{K}} + \frac{1}{K} + \frac{1}{K\sqrt{K}})$. $\qquad \square$

### A.7.2 PROOF OF COROLLARY 2.1

Define the cumulative reward function of the expert as $J_e(\pi) \triangleq E^\pi [\sum_{t=0}^{T-1} \gamma^t r_{\theta_e}(s^t, a^t)]$. If $r_{\theta_e}$ is a linear reward function, we have $r_e^{\theta_e} \triangleq \langle \theta_e, \phi(s, a_e) \rangle$ where the feature $\phi(s, a_e) \in \mathbb{R}^{d_{\theta_e}}$ and $d_{\theta_e}$ is the dimension of $\theta_e$. Then the expert's feature expectation is formulated as $\mu_f(\pi) \triangleq E^\pi [\sum_{t=0}^{T-1} \gamma^t \phi(s^t, a_e^t)]$. From Theorem 2, we can get $\|\mu_f(\pi_{\theta_e}) - \mu_f(\pi_e)\|^2$ decreases in $\mathcal{O}(\frac{1}{\sqrt{K}})$.

*Proof.*

$$J_e(\pi_{\theta_e}) - J_e(\pi_e) = \langle \theta_e, \mu_f(\pi_{\theta_e}) - \mu_f(\pi_e) \rangle$$
$$\leq \max_{\theta_e \in \Theta} \langle \theta_e, \mu_f(\pi_{\theta_e}) - \mu_f(\pi_e) \rangle$$
$$\overset{(v)}{\leq} \max_{\theta_e \in \Theta} \|\theta_e\| \|\mu_f(\pi_{\theta_e}) - \mu_f(\pi_e)\|,$$
$$= \|\mu_f(\pi_{\theta_e}) - \mu_f(\pi_e)\|,$$
$$\overset{(vii)}{\leq} \frac{C_{15}}{\sqrt[4]{K}},$$

where $(vii)$ uses the result $\|\mu_f(\pi_{\theta_e}) - \mu_f(\pi_e)\|^2$ decreases in $\mathcal{O}(\frac{1}{\sqrt{K}})$ and $C_{15}$ is the constant number which includes all influence factors other than $K$. $\qquad \square$

### A.8 PROOF OF THEOREM 1

The prove of Theorem 1 is as follows: According to the Algorithm 1 in (Ziebart et al., 2008), we need to compute the state frequency for the $\mu_e(\pi_{\theta_l, \theta_e})$. For each state-action pair, it needs to recursively compute for up to $T$ iterations. As there are $T$ state-action

pairs in one demonstration, the computational complexity for the $\mu_e(\pi_{\theta_l, \theta_e})$ is $\mathcal{O}(T^2)$ as we see $T$ as the deterministic factor. Analogously, the computational complexity for $\mu_l(\pi_{\theta_l, \theta_e}), \mu_e(s, a), \mu_e(s), \mu_l(s, a), \mu_l(s), J_{ei}(s, a), J_{ed}(\pi_{\theta_l, \theta_e}), J_{ei}(\pi_{\theta_l, \theta_e})$ are $\mathcal{O}(T^2)$. We can see that these factors with computational complexity $\mathcal{O}(T^2)$ are expected to be calculated through backpropagation, and the backpropagation leads to $\mathcal{O}(T^2)$. However, in coding, we can calculate these factors by sampling a finite number of trajectories, and the computational complexity becomes $\mathcal{O}(T)$.

For SPSA, the required terms are $f(\theta_l, \theta_e), \nabla_{\theta_e} L(\theta_l, \theta_e), \nabla_{\theta_l} L(\theta_l, \theta_e)$, these terms are the linear combination of $\mathcal{O}(T)$ computational complexity terms, so their computational complexities are also $\mathcal{O}(T)$.

If we directly compute $\nabla_{\theta_l} f(\theta_l(k), \theta_e(k))$, $\nabla^2_{\theta_l \theta_e} L(\theta_l(k), \theta_e(k))$, $\nabla^2_{\theta_e} L(\theta_l(k), \theta_e(k))$, and $\nabla_{\theta_e} f(\theta_l(k), \theta_e(k))$ instead of approximating, use the expression of $\nabla_{\theta_l} f(\theta_l, \theta_e) = E^{\pi_{\theta_l, \theta_e}} [\sum_{t=0}^{T-1} \gamma^t [(\mu_l(s^t, a^t) - \mu_l(s^t))(J_{ed}(s^t, a^t) + J_{ei}(s^t, a^t))^T]] + \mu_l(\pi_{\theta_l, \theta_e})$ as an example, $\mu_l(s)$ is inside an expectation from $t = 0$ to $T - 1$, we need to sum up $\mu_l(s)$ for $T$ times. As a result, the computational complexity for $\nabla_{\theta_l} f(\theta_l, \theta_e)$ is $\mathcal{O}(T^2)$. Analogously, $\nabla_{\theta_e} f(\theta_l, \theta_e)$, $\nabla^2_{\theta_l \theta_e} L(\theta_l, \theta_e), \nabla^2_{\theta_e \theta_e} L(\theta_l, \theta_e)$ are all same.

Back to SPSA, more policies need to be found compared to directly compute the hypergradient. Use soft q learning (Haarnoja et al., 2017) as an example, RL and MARL algorithms can be considered as sampling a finite number trajectories for each epoch. Therefore the computational cost for each epoch is $\mathcal{O}(T)$ and the overall computational cost is $\mathcal{O}(eT)$ where $e$ is the total number of epochs. As we consider $T$ as the decision factor, the computational cost of the multi-agent RL is $\mathcal{O}(T)$, which matches the computational complexity of SPSA.

As a result, the computational complexity of SPSA is dominated by $\mathcal{O}(T)$, and directly calculating the hypergradient is dominated by $\mathcal{O}(T^2)$.

## A.9 Experiment Detail

The details of the experiments are shown in this section. All Python3 codes are run on a Windows 10 desktop with 13th Gen Intel(R) Core(TM) i7-13700KF CPU and 32 GB of RAM. For each combination of algorithms and environments, we run 10 times to calculate mean values and standard deviations at each iteration. Then the calculated mean values and standard deviations are plotted as shown in Section 7 figures.

### A.9.1 MPE

The state, action, and observation spaces for the adversary and good agents are continuous. For the adversary, it can observe the relative distance to the landmarks and the good agents, therefore the observation of the adversary is $o_a = [p_{l1} - p_a, p_{l2} - p_a, p_{g1} - p_a, p_{g2} - p_a]$ where $p_{l1}$ is the position of the first landmark, $p_{l2}$ is the position of the second landmark, $p_{g1}$ is the position of the first good agent, $p_{g2}$ is the position of the second good agent. For each good agent, it can observe the relative distance to the target landmark, the landmarks, the adversary, and another good agent, therefore the observations for two good agents are $o_{g1} = [p_{tl} - p_{g1}, p_{l1} - p_{g1}, p_{l2} - p_{g1}, p_a - p_{g1}, p_{g2} - p_{g1}]$ and $o_{g2} = [p_{tl} - p_{g2}, p_{l1} - p_{g2}, p_{l2} - p_{g2}, p_a - p_{g2}, p_{g1} - p_{g2}]$ where $p_{tl}$ is the position of the target landmark. The actions of the adversary and the good agents are the velocities between $0$ and $1$ in four directions (left, right, down, up). Two good agents share the same return, which is rewarded based on the minimum distance of any agent to the target landmark and is penalized based on the distance between the adversary and the target landmark, therefore the reward of good agents is $r_g = -\min(\|p_{tl} - p_{g1}\|_2, \|p_{tl} - p_{g2}\|_2) + \|p_{tl} - p_a\|_2$. The reward of the adversary is based on the distance to the target of the adversary, therefore $r_a = -\|p_{ta} - p_a\|_2$, where $p_{ta}$ is the position of the adversary's target. In our simulation, we consider observations as states of the MG, the distance $-\min(\|p_{tl} - p_{g1}\|_2, \|p_{t2} - p_{g1}\|_2)$ as the adversary-independent reward function, and the distance $\|p_{tl} - p_a\|_2$ as the feedback received by the good agents.

### A.9.2 SMAC

In this scenario, we can only access the state, observation, and action information of agents. The enemy is controlled by a built-in game AI. We consider observations as the states of the agents during learning. For each agent, it can observe the following information corresponding to another agent and enemy: relative distance, relative x, relative y, health, shield, and unit type. If an agent or an enemy is under attack, the shield reduces first and health reduces after the shield disappears. There are 7 possible actions for each agent: move north, move south, move east, move west, attack the enemy, stop, and no-op. At each time step, each agent can know which action among these 7 possible actions are available and the agent chooses one action from available actions. For example, the attack action is available when the enemy is in the shooting range of the agent. The agent can only choose no-op when its own health is 0. The agent gains rewards based on the damage dealt to the enemy and if the enemy is defeated.

### A.9.3 HUMAN-ROBOT INTERACTION

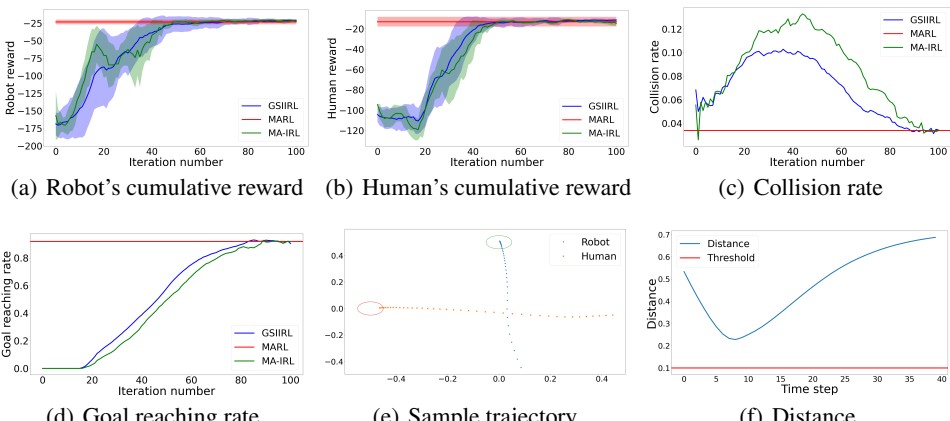

(a) Robot's cumulative reward  (b) Human's cumulative reward  (c) Collision rate

(d) Goal reaching rate  (e) Sample trajectory  (f) Distance

Figure 4: Human–robot interaction simulation results. The cumulative rewards are calculated as described in Figure 2. Similarly, the collision and goal-reaching rates are computed based on the policies corresponding to the learned reward functions. A collision is defined to occur when the distance between the human and the robot falls below a threshold. The goal is considered reached only when both the human and the robot reach their respective destinations.

Consider the human-robot interaction scenario described in the introduction, both the robot (learner) and the human (expert) aim to reach their destinations as quickly as possible. Additionally, the robot strives to maintain the human's right-of-way. Both agents operate in continuous state and action spaces: each has a 4-dimensional state space and a 2-dimensional action space.

Figures 4(a) and 4(b) show that the robot's and the human's cumulative rewards converge to the ground-truth values, consistent with previous experiments. Figure 4(c) shows that the collision rate converges to its ground-truth value, indicating that the robot successfully learns a human-dependent reward function. Initially, with randomly generated reward functions, the robot and human move randomly, and collisions are rare. As the reward functions are iteratively updated, both agents move toward their goal locations, causing collisions and an increase in the collision rate. However, as training continues and the reward functions improve to prioritize collision avoidance, the collision rate decreases. Figure 4(d) shows that the goal-reaching rate also converges to the ground-truth value. With the learned reward functions, both agents reliably reach their goals. After 100 iterations of reward function updates, the learned robot reward function yields a high goal-reaching probability and a low collision probability. Figure 4(e) presents a sample trajectory under the learned reward functions, and Figure 4(f) plots the distance between the robot and the human along this trajectory. Both agents reach their goals while maintaining a separation above the required safety distance.

In the simulation, the policies for the human and the robot are calculated through Multi-Agent Deep Deterministic Policy Gradient (MADDPG) (Lowe et al., 2017). During the training process, noises are added to the action to increase the exploration. Once the policies are generated, further calculations use deterministic policies.

The state of the robot is its location $s_r = (x_r, y_r) \in \mathbb{R}^2$, the action of the robot includes the horizontal and vertical velocities and defined as $a_r = (v_{rx}, v_{ry})$, where $v_{rx} \in [-0.1, 0.1], v_{ry} \in [-0.1, 0.1]$. Similarly, the state of the human is $s_h = (x_h, y_h) \in \mathbb{R}^2$, the action of the human is $a_h = (v_{hx}, v_{hy})$, where $v_{hx} \in [-0.1, 0.1], v_{hy} \in [-0.1, 0.1]$. At each time step $t$, the robot chooses the action $a_r(t)$ based on the current joint state $(s_r(t), s_h(t))$ and moves to the next state $x_r(t+1) = x_r(t) + v_{rx}(t), y_r(t+1) = y_r(t) + v_{ry}(t)$. Analogously, the motion dynamics of the human is given by $x_h(t + 1) = x_h(t) + v_{hx}(t), y_h(t + 1) = y_h(t) + v_{hy}(t)$. In the experiment, the robot starts from an initial state $s_r(0) \in [-0.25, 0.25] \times [-0.4, 0.5]$ and aims to reach a circle goal region whose center is at $(0, 0.5)$ with radius $0.05$. The human starts from $s_h(0) \in [0.4, 0.5] \times [-0.25, 0.25]$ and aims to reach a circle goal region whose center is at $(-0.5, 0)$ with radius $0.05$. Both the robot and the human are penalized when a collision happens.

### A.9.4 SECURITY

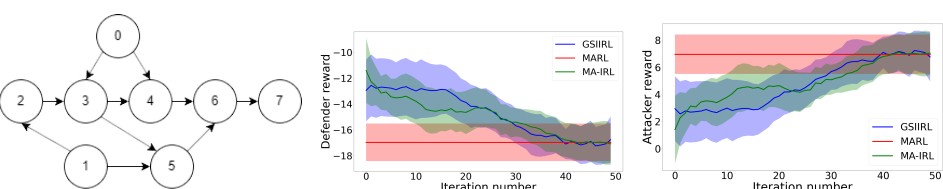

Figure 5: Cyber security simulation results. **Left**: Attack graph. **Middle**: Defender's reward. **Right**: Attacker's reward. The cumulative rewards are calculated in the same way described in Figure 2.

We conduct a cybersecurity experiment to evaluate the proposed algorithm. The experimental setup involves a defender (learner) and an attacker (expert) interacting on the attack graph in Figure 5. The attacker attempts to compromise nodes in the graph. The attacker's objective is to compromise as many nodes as possible, whereas the defender's objective is to protect the network by minimizing the number of compromised nodes. Both the learner and the expert have discrete state and action spaces, with cardinalities of $256$ and $8$, respectively.

Figure 5 shows that the cumulative rewards of the proposed algorithm converge to those of MARL, consistent with previous experiments.

There are $8$ nodes and $10$ edges. Each node represents a machine, and each edge represents an exploit between two nodes. The decision-making of the defender and the attacker is modeled as an MG. The state $s \in \{0, 1\}^8$, represents the condition of each node where the value $1$ means the current node is compromised by the attacker, and the value $0$ is vice versa. In each action pair, the attacker chooses one edge to attack, and the defender chooses one edge to block. Suppose the attacker chooses to attack the edge $\{i, j\}$. If the node $i$ is already compromised and the defender does not block this edge, there is a probability for node $j$ to be compromised. For other situations, the node $j$ keeps clean. Each edge has a cost for the attacker to utilize and a cost for the defender to block. The attacker receives a reward when it successfully compromises a new node. The net reward of the attacker for each state-action pair is the sum of the reward and the cost. For the defender, the expert-dependent reward is the opposite of the attacker's reward, and the expert-independent reward is the cost to block edges.

For the security simulation, the attack graph is randomly generated. We use Q-learning to find the policies for the attacker and the defender. During the training process, the attacker and the defender have a $70\%$ possibility to choose between the best action with $60\%$ possibility and the second best action with $40\%$ possibility. Otherwise, the attacker and the defender randomly choose one action from the action space. When we exploit the learned policies, the attacker and the defender choose between the best action with $60\%$ possibility and the second best action with $40\%$ possibility.

### A.9.5 CUMULATIVE REWARDS COMPARE TABLE

### A.10 THE USE OF LARGE LANGUAGE MODELS (LLMS)

We confirm that LLM (ChatGPT 5) assistance was limited to improving grammar and readability.

Table 1: The cumulative reward comparison. **Top**: The cumulative reward of the learner. **Bottom**: The cumulative reward of the expert. MARL uses ground-truth reward functions. MA-IRL and GSIIRL use learned reward functions from the last iteration.

| LEARNER | MARL | MA-IRL | GSIIRL |
|---|---|---|---|
| MPE | $5.03 \pm 1.66$ | $5.12 \pm 4.08$ | $4.84 \pm 1.81$ |
| SMAC | $19.01 \pm 0.13$ | $18.89 \pm 0.20$ | $18.87 \pm 0.27$ |
| CS | $-16.96 \pm 1.45$ | $-17.08 \pm 3.21$ | $-17.38 \pm 3.59$ |
| HRI | $-20.22 \pm 3.34$ | $-20.63 \pm 1.07$ | $-21.09 \pm 1.87$ |
| EXPERT | MARL | MA-IRL | GSIIRL |
| MPE | $-18.91 \pm 2.01$ | $-20.37 \pm 4.27$ | $-19.53 \pm 2.50$ |
| SMAC | $19.01 \pm 0.13$ | $18.89 \pm 0.20$ | $18.87 \pm 0.27$ |
| CS | $6.96 \pm 1.45$ | $7.03 \pm 3.21$ | $6.76 \pm 3.59$ |
| HRI | $-12.98 \pm 4.78$ | $-13.52 \pm 2.57$ | $-13.16 \pm 1.03$ |

