# OpenReview forum: "Inverse Reinforcement Learning of Interactive Scenarios"
_ICLR.cc/2026/Conference — Submitted to ICLR 2026_

### Official Review · Reviewer_L9NX · 2025-10-21

**Soundness:** 1
**Presentation:** 1
**Contribution:** 1
**Rating:** 2
**Confidence:** 5

**Summary:**

This paper is about IRL in settings where the learner actively interacts with the expert and can affect its behavior.

**Strengths:**

/

**Weaknesses:**

The problem formulation is written in a very confusing manner so that it is not possible to understand:
1) what is the problem that this paper aims to solve;
2) what is the mathematical model that it considers.

To be more precise, in lines 118-139 the authors describe a markov game with two agents, the learner and the expert, with their own reward functions that they aim to optimize. Ok, this is fine.

But then, lines 140-143: "the learner aims to recover the expert’s reward function $r\_e$ and compute the optimal joint policy $\pi^\*$". What does it mean? No notion of joint optimal policy $\pi^*$ was defined earlier, not even a notion of equilibrium. I assume that the goal is to find the learner optimal policy, i.e., the policy $\pi_l$ that maximizes the expected return of the learner w.r.t. the best possible policy that the expert can play. However, there is another problem here. The paper tells us that the learner reward $r_{ld}$ is unknown. What does it mean? I mean, if the expert's reward $r_e$ is unknown is fine, because the expert policy gives us information about it. But if $r_{ld}$ is unknown, we do not have any way to have information about it! Indeed, the learner observes only state-action trajectories.

Lines 144-182 do not provide any information about this issue. Instead, Section 4 makes things even more confusing. Here, the authors introduce other "joint optimal policies" $\pi_{\theta_l,r_e}$, without defining them formally. Moreover, it seems from this section that the learner can choose a reward $\theta_l$ and then it can observe trajectories from the "optimal policy" $\pi_{\theta_l,r_e}$. Thus, here, it seems that the "joint optimal policy" is obtained regardless of the policy played by the other player, but this does not make sense, as the whole goal is about the interaction.

**Questions:**

clarify the problem setting

**Details Of Ethics Concerns:**

I reviewed this paper also for ICML 2025. It was rejected because the problem setting was completely unclear. I expected this submission to ICLR to be much better, but there are just very very minor adjustments in the presentation (e.g., adding few words of explanation instead of rewriting entire sections).

---

### Official Review · Reviewer_pH7L · 2025-10-22

**Soundness:** 2
**Presentation:** 1
**Contribution:** 2
**Rating:** 2
**Confidence:** 2

**Summary:**

The paper studies a non-cooperative multi-agent problem in which the learner aims to maximize a reward function composed of a learner-centric reward and an expert-dependent reward that is unknown to the agent. To achieve this, the learner must act in a shared environment with the expert, who is assumed to behave optimally with respect to an expert reward (which may depend on the learner's behavior). The expert reward is unknown to the learner. To address this problem, the authors propose to infer both the expert reward function and the optimal joint policy. This leads to a bi-level optimization problem, as the two problems are intertwined.

**Strengths:**

- The problem is novel as far as I know and interesting but I'm not very familiar with the MARL literature
- The authors provides a detailed comparison with both the IRL and the MARL literature
- The proposed solution is elegant and seems sound (I did not have the time to review all the appendix)

**Weaknesses:**

- The paper is not clear

**Questions:**

- From the paper, it is unclear what the learner can and cannot observe during its interaction with the environment. Does the learner observe the state of the expert? Does it receive the reward $r_{ld} + r_{li}$? I assume that $r_{ld}$ is not observable; otherwise, this would reduce to a standard RL problem where the expert is simply a non-controllable component of the environment. Specifically, in Equation 2, where does $\theta^\star_e$ play a role?

- But if $r_{ld}$ is not observable, how do you ground the estimate of $r_{ld}$? I may have missed something, but it seems that the problem is not well posed. Abstracting from the concrete problem, since the expert is not directly controlled, I could always maximize (2) by making the estimated reward $r_{\theta_l} \to \infty$. Taking inspiration from the human-robot interaction example in your paper: if I do not know the expert-dependent component of the reward and I try to learn it by maximizing the reward, I could find degenerate solutions that dominate the $r_{li}$ reward. Are you avoiding this by assuming a bound on the reward function (even though this is not stated)? Or are you simply bounding the parameters?

At the moment, my score is negative, but I may have missed important aspects (I indicated this by using a low confidence score). Looking forward to receiving clarifications from the authors about these two fundamental questions.

---

### Official Review · Reviewer_7UsQ · 2025-10-25

**Soundness:** 3
**Presentation:** 2
**Contribution:** 2
**Rating:** 4
**Confidence:** 3

**Summary:**

This paper considers a novel IRL setting where the learner can interact with the expert and the learning has two components of reward functions, one dependent on the expert and one independent of the expert. The problem is formulated as a bi-level optimization where the lower level learns the expert reward function, the upper level optimizes its own combined reward. The authors use SPSA to approximate the hyper-gradient, bypassing expensive Jacobian and Hessian computation. Experiments are conducted on 4 environments. It was shown that the proposed method matches the performance of MARL but require less information (expert reward), while other methods can performance much worse.

**Strengths:**

* The problem setting of interactive IRL is novel to my knowledge and generalized fully cooperative or adversarial IRL.
* The problem formulation and algorithm generally make sense.
* The experimental results are convincing and address the main research question for this special setting. Using MARL as a benchmark makes the interpretation of results especially clear and easy.

**Weaknesses:**

* The problem formulation in section 4 is not very well presented. A part of this might be due to overloaded notations.

**Questions:**

* In equation 2, the objective is to optimize the learner policy w.r.t. the learner combined reward right? On line 190, you used $\theta_{l}$ to denote the parameters of expert dependent reward $r_{ld}$. But in Eq. 2, you are minimizing w.r.t. $\theta_{l}$, so not the policy?
* Does the expert know the learner's reward functions, both expert dependent and independent parts? The assumption in the likelihood in Eq. 1 seems to be that the expert knows the learner's reward, such that the gradient in Lemma 1 can be obtained. If so this seems to be a strong assumption.
* Can you provide some intuition for the bi-level optimization in Eq. 3? I understand that the bi-level structure comes from the fact that estimating expert reward parameters depends on the learner reward. But I have a suspicion that the bi-level structure induces some form of recursive reasoning.

---

### Official Review · Reviewer_arXG · 2025-10-29

**Soundness:** 1
**Presentation:** 1
**Contribution:** 1
**Rating:** 2
**Confidence:** 4

**Summary:**

The paper investigates the inverse reinforcement learning (IRL) problem in interactive settings, where the learner not only needs to infer the expert’s reward function but also affects the expert’s behavior through its own policy. To address this, the authors propose a bi-level optimization framework and develop the GSIIRL algorithm. GSIIRL alternates between learning the expert’s reward function (inner loop) and learn the joint policy (outer loop). The authors reduce the computational burden by using SPSA techniques and prove that it converges at a rate of $O(1/\sqrt{K})$. Experimental results on multiple multi-agent environments show that GSIIRL achieves performance comparable to MARL methods that have access to the ground-truth rewards.

**Strengths:**

1. Unlike prior methods that assume either fully cooperative or fully competitive, the paper intends to model a more general relationship between the expert and the learner.
2. The authors develop an optimization procedure based on SPSA to efficiently approximate gradients and reduce the computational cost of the original bi-level optimization problem
3. The authors provide a provable convergence rate of $O(1/\sqrt{K})$, matching the best known rates for bilevel optimization methods
4. The algorithm converges across various multi-agent RL settings and is compared to properly designed baselines

**Weaknesses:**

1. My main concern lies in the formulation of the interactive IRL problem. As I understand it, the expert’s reward function $r_e$ is completely unknown, while the expert-dependent component of the learner’s reward $r_{ld}$ is also unknown but can be observed when learning the joint policy $\pi^*$ (Based on Equation 2). This raises a few issues
    * If $r_{ld}$ is observable during this process, how is it different from $r_{li}$? Is the functional form of $r_{li}$ fully known? In this case, it is unclear why an IRL framework is necessary. Would it not be more straightforward to approximate $r_{ld}$ with a parametric model and learn the joint policy $\pi_{\theta_l,\theta_e^*}$ using model-based reinforcement learning? In contrast, applying IRL to recover $r_e$ seems well justified, as the expert’s reward signals remain unobserved.
    * Although previous studies focus on fully cooperative or fully competitive scenarios, they generally do not assume prior knowledge of the learner’s reward function.

2. I could not find an explicit description—either in Algorithm 1 or in the Appendix—of how the joint policy $\pi_{\theta_l,\theta_e^*}$ is actually learned. Is the policy optimization process model-based or model-free? Clarifying this would strengthen the paper’s technical transparency and reproducibility.

**Questions:**

1. The definition of $f(\theta_,\theta_e^*(\theta))$ in Equation 2 seems to differ from that in Appendix A.1. Specifically, Equation 2 uses $r_{ld}+r_{li}$ while $J_{ed}$ uses $r_{\theta_l}$ instead.

2. What is $p(k)$ in Equation 4? Is it a constant or an iteration-dependent parameter?

3. The results for CIRL and ML-IRL baselines are not shown in Figures 4 and 5.

4. The paper does not provide enough details about the multi-agent RL (MARL) baseline. How is the MARL reward function defined? Does it use the sum of individual agent rewards or another aggregation scheme? If the sum is used, then the paper should also report the sum of rewards over iterations to properly assess convergence.

5. In the proof of Theorem 1, it is stated that computing $\triangledown_{\theta_e} L$ only requires O(T), even though it involves evaluating $\mu_e(\pi_{\theta_l,\theta_e})$ (Lemma 1), which is described earlier as having $O(T^2)$. This seems inconsistent

---

### Meta-Review · Area_Chair_kSd2 · 2026-01-05

**Summary:**

The reviewers unanimously recommended the rejection of the paper based on concerns and open questions regarding the problem formulation and the capabilities of the involved agents. As the authors did not provide a rebuttal, the questions remain unanswered and concerns unrefuted.

**Reviewer Concerns:**

No rebuttal was provided, so all concerns are still standing.

**Reviewer Scores:**

n/a (no discussion would haven taken place because no rebuttal was submitted)

---

### Decision · Program_Chairs · 2026-01-26

Reject